# Samba: Severity-aware Recurrent Modeling for Cross-domain Medical Image Grading

**Qi Bi**[1,*] **Jingjun Yi**[2], **Hao Zheng**[2✉], **Wei Ji**[3], **Haolan Zhan**[4],
**Yawen Huang**[2], **Yuexiang Li**[5✉], **Yefeng Zheng**[1✉]
[1]Westlake University, China, [2]Jarvis Research Center, Tencent Youtu Lab, China,
[3]Yale University, United States, [4]Monash University, Australia,
[5]Guangxi Medical University, China
howzheng@tencent.com, yuexiang.li@ieee.org
zhengyefeng@westlake.edu.cn

## Abstract

Disease grading is a crucial task in medical image analysis. Due to the continuous progression of diseases, *i.e.*, the variability within the same level and the similarity between adjacent stages, accurate grading is highly challenging. Furthermore, in real-world scenarios, models trained on limited source domain datasets should also be capable of handling data from unseen target domains. Due to the cross-domain variants, the feature distribution between source and unseen target domains can be dramatically different, leading to a substantial decrease in model performance. To address these challenges in cross-domain disease grading, we propose a Severity-aware Recurrent Modeling (Samba) method in this paper. As the core objective of most staging tasks is to identify the most severe lesions, which may only occupy a small portion of the image, we propose to encode image patches in a sequential and recurrent manner. Specifically, a state space model is tailored to store and transport the severity information by hidden states. Moreover, to mitigate the impact of cross-domain variants, an Expectation-Maximization (EM) based state recalibration mechanism is designed to map the patch embeddings into a more compact space. We model the feature distributions of different lesions through the Gaussian Mixture Model (GMM) and reconstruct the intermediate features based on learnable severity bases. Extensive experiments show the proposed Samba outperforms the VMamba baseline by an average accuracy of 23.5%, 5.6% and 4.1% on the cross-domain grading of fatigue fracture, breast cancer and diabetic retinopathy, respectively. Source code is available at `https://github.com/BiQiWHU/Samba`.

## 1 Introduction

Disease grading aims to assess the severity level of a disease or a pathological region from a medical image [46, 31, 52, 50, 6]. It is more challenging than conventional deterministic classification with distinctive categories (*e.g.*, cat *vs.* dog), owing to the inherent severity ambiguity within and between levels. This ambiguity arises because the progression of a certain disease or a pathological region is a transitional, continuous and time-growing process (illustrated in Fig. 1a). On the one hand, different medical images within a same severity level can have rather different disease or pathological developments (shown in Fig. 1b). On the other hand, medical images among different severity levels can share similar patterns, as low-level lesions may persist throughout the disease's progression.

---

[*]Qi Bi is affiliated with University of Amsterdam. This research was conducted with Westlake University and Tencent Youtu Lab.

38th Conference on Neural Information Processing Systems (NeurIPS 2024).

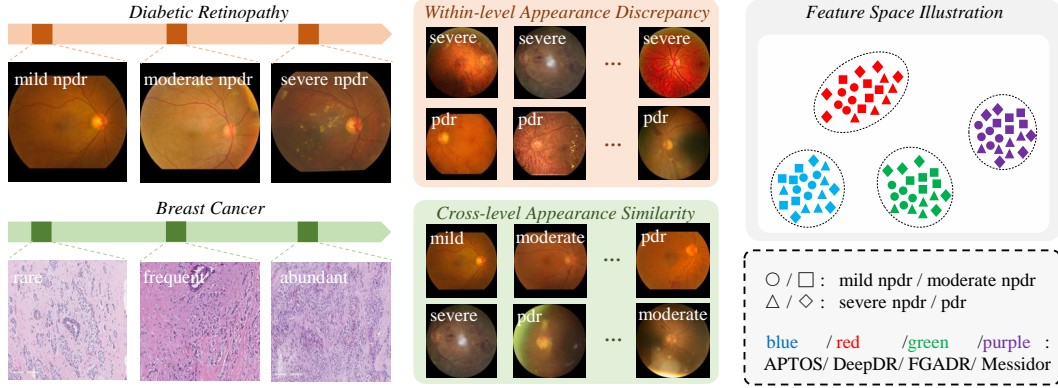

Figure 1: (a) The development of a disease or a pathological region is a continuous progress; (b) The continuous development apart from individual differences and style variation poses both within-level discrepancy (top) and cross-level similarity (bottom) on the medical image appearance; (c) These challenges can make the medical images from the same unseen domain, instead of those from the same grade level, to be clustered in the feature space.

The past decade has witnessed the rapid development of disease grading methods [30, 34, 45] owing to the deep learning techniques [25, 24, 28, 26, 57]. However, most of these methods were developed by experts from a specific clinic field (ophthalmology, gynecology, *etc.*). Furthermore, these models usually assume that the medical images used for training and inference are independently and identically distributed (i.i.d.). In practical clinical scenarios, a grading model trained on a number of medical images (source domain) is often required to handle images it has not encountered before (unseen target domains). Due to variations between patients, scanners, imaging parameters, clinic centers, *etc.*, the feature distributions of the source domain and unseen target domains can be dramatically different [32, 68, 23, 8]. When the severity level of a disease is measured by the distribution of lesions, this cross-domain variance can lead to the misdetection of crucial lesions, resulting in grading errors [4, 11]. Especially when the appearance of lesions is significantly affected by the style change, it may be observed that medical images from the same *domain* instead of from the same *grade* are clustered in the feature space (illustrated in Fig. 1c). This suggests that the model has learned features with limited generalization ability.

Domain generalized disease grading learns models from only a source domain, but is expected to be applicable to unseen target domains. The key to addressing this problem is accurately identifying the lesions that have a decisive impact on grading [49, 45]. As the disease progresses, multiple lesions may coexist in the image, and the critical aspect of grading is identifying the most severe one among them. However, the most severe lesion may be localized in a small region in the image, exhibiting variable shapes, and being influenced by cross-domain style changes [4, 11]. To overcome these challenges, this paper proposes a severity-aware recurrent modeling method (Samba). Samba encodes image patches in a recurrent manner and recalibrates the state distributions based on learnable bases.

In many disease grading scenarios, the decisive lesions only occupy a small portion of the total area. For instance, in retinal photographs, the affected blood vessels may only involve a small section at the distal. Similarly, in computed tomography (CT) or magnetic resonance imaging (MRI) scans, malignant tumors can also present as small lesions with a diameter less than 3 mm. These small lesions are easily influenced by style variations, which can lead to incorrect grading. Therefore, the model needs to pay sufficient attention to these detailed patches to classify them accurately. To address this issue, we treat the image patches as sequential data and encode them in a recurrent manner. This approach allows the information of decisive lesions to be stored in the hidden states and propagated to subsequent sequences. Furthermore, we adopt bidirectional encoding, enabling critical local information to influence the overall representation. More specifically, we incorporate a bidirectional Mamba [17] layer into the Samba, which supports sequence-to-sequence transformation and efficiently selects data in an input-dependent manner.

The Mamba model achieves its selection mechanism by parameterizing the State Space Model (SSM) based on the input. While this selection mechanism [17, 69, 35] aids in identifying decisive lesions

and propagating critical information, these input-dependent parameters are also vulnerable to the influence of image style transformations. When the feature distribution is affected by cross-domain variations, both the update of hidden states and the gating mechanism are disrupted. To resolve this problem, we utilize learnable tokens to capture the lesion representations, which are then used as bases to map the feature embeddings into a more compact space. To preserve the semantic information within this process, we further employ the Expectation-Maximum (EM) algorithm [14] initialized by these bases to estimate the lesion feature distribution for each image and reconstruct the features accordingly. We refer to this process as EM-based state recalibration in this paper.

Our contributions can be summarized as follows.

- We develop a Severity-aware Recurrent Modeling, dubbed as Samba, for general disease grading within- and cross-domain medical images.
- We propose to encode the image patches in a recurrent manner to accurately capture the decisive lesions and transport the critical information from local to global.
- An EM-based state recalibration mechanism is designed to reduce the impacts of cross-domain variants by mapping the feature embeddings into a compact space.
- Extensive experiments on three cross-domain disease grading benchmarks show the effectiveness of the Samba against the baseline.

## 2 Related Work

**Domain generalization** aims to learn a model that can be generalized to unseen target domains when only trained by the source domain, where the cross-domain feature distribution is usually not identical [65]. A variety of machine learning techniques (*e.g.*, discrepancy minimization [47, 13], knowledge ensemble [12], uncertainty quantification [39, 53], optimal transport [16, 60], self-learning [51, 43], frequency decoupling [59, 9, 10] and casual inference [37, 38]) have been proposed. In the medical imaging community, the effort of bridging the domain gap between training data and unseen inference data is so far mainly focused on medical image segmentation [32, 68, 23, 8, 58] and classification [66, 54]. These methods usually rely on either learning shape-invariant representation or reaching pixel-wise consensus among the source domains. However, they are not especially devised to tackle the key challenge in cross-domain medical image grading, where the medical images from the same severity level instead of the same domain tend to cluster together.

**Medical grading** has also been studied. For Diabetic Retinopathy (DR) grading, many works highlight the subtle local pathological regions to better discern different severity levels [30, 34, 44, 6, 45, 7]. Similarly, grading models have also been developed for pulmonary nodules [46], fatigue fracture [31], glioma [52], acne vulgaris [50], *etc*. However, most of the existing grading methods are task-specific and assume the training and inference medical images are i.i.d., which is far from reality. Practically, a medical grading model is supposed to show reliable inference on unseen target domains that have different feature distribution from the source domain. *To the best of our knowledge*, only [4] and [11] made an initial investigation on learning domain generalized DR grading.

**State Space Model** (SSM) [27] contributes to a variety of fields such as robotics, navigation, and control theory, which is a foundational scientific model. In the past few years, SSM has been adapted in the context of deep representation learning, and has shown great success in sequence modeling [19, 20]. More advanced SSM, exemplified by Mamba [17], not only shows stronger representation ability in long sequence modeling, but also exhibits linear scaling ability for long-sequence data. Built upon this, multiple Mamba variations (*e.g.*, Vim [35] and VMamba [69]) have shown effectiveness in the computer vision field. However, these methods mainly focus on enhancing the context representation from the image by exploiting the long-range dependencies. Instead, how to model the cross-level severity development from the medical image by SSM remains unexplored.

## 3 Methodology

### 3.1 Problem Definition & Framework Overview

For a given disease grading task, assume we have a number of medical images $x$ and the corresponding severity-level labels $y$ from $K$ different domains, which is denoted as $\mathcal{D}_1 = \{(x_n^{(1)}, y_n^{(1)})\}_{n=1}^{N_1}$,

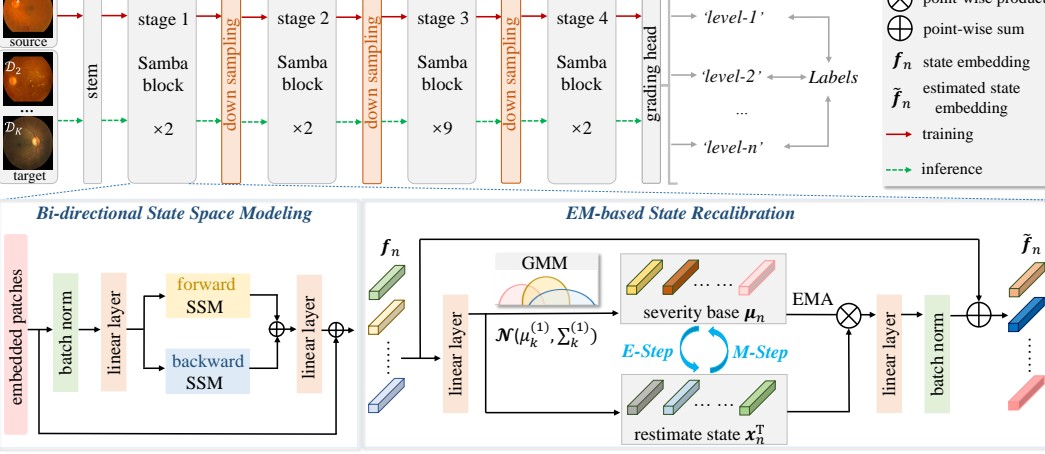

Figure 2: Framework of the proposed Severity-aware Recurrent Modeling (Samba) method. The patch embeddings pass through four encoding stages consisting of different number of severity-aware recurrent layers. Within each Samba block, the embeddings are first input to the bidirectional Mamba layers to store and transport the information about decisive lesions. After that, an EM-based state recalibration module models the feature distribution of lesions via a Gaussian Mixture Model with learnable severity bases. Moreover, the bases are re-estimated by the EM algorithm for each image and reconstruct the features finally.

$\mathcal{D}_2 = \{(x_n^{(2)}, y_n^{(2)})\}_{n=1}^{N_2}, \cdots, \mathcal{D}_K = \{(x_n^{(K)}, y_n^{(K)})\}_{n=1}^{N_K}$. Here $N_k$ denotes the number of images in domain $k$. For the cross-domain disease grading problem, the objective is to learn a grading model $F_\theta : x \to y$ using images only from a source domain $\mathcal{D}_1$, which is supposed to generalize well on other unseen target domains $\mathcal{D}_2, \cdots, \mathcal{D}_K$. Following prior domain generalization works, each dataset is regarded as a domain $\mathcal{D}_k$, as the samples in a certain dataset are usually collected from the same clinical center by the same scanners and therefore share more similar feature distribution.

The overview of the proposed method is illustrated in Fig. 2. The input image is first encoded into patch embeddings through a stem unit with $4 \times 4$ convolutional kernels, where the stem unit partitions the input image into patches. The patch embeddings further pass through four encoding stages. Each Samba block involves a certain number of severity-aware recurrent layers and there are downsampling layers between two consecutive blocks. Finally, a grading head consisting of an average pooling layer and a linear layer generates the final prediction. Within the Samba block, the patch embeddings are first input to the bidirectional Mamba layers to extract the information about decisive lesions. After that, EM-based state recalibration is applied to map the lesion representation into a compact space by learnable bases.

## 3.2 Recurrent Patch Modeling by State Space Model

The core issue in most medical image disease grading scenarios is to identify the most severe lesion. However, due to the presence of lesions from different stages of the disease in the image, accurately capturing the most severe lesion is highly challenging. When the lesion occupies a large proportion of the image, the model only needs to extract stage-related features. In contrast, when the area of the critical lesion is small, the model needs to simultaneously locate the lesion and extract relevant features. This places higher demands on the model's ability to handle local information. To address this issue, in this paper, we propose to encode the image patches in a recurrent manner. Specifically, the state space model is used to process the sequential patch embeddings.

**State Space Model.** Let $x(t)$ denote a 1-D input signal. SSM maps it to the 1-D output signal $y(t)$ by an intermediate $N$-dimensional latent state $u(t)$, given by

$$u'(t) = \boldsymbol{A}u(t) + \boldsymbol{B}x(t), \quad y(t) = \boldsymbol{C}u(t) + \boldsymbol{D}x(t), \tag{1}$$

where $\boldsymbol{A} \in \mathbb{R}^{N \times N}$ denotes the state matrix, while $\boldsymbol{B} \in \mathbb{R}^{N \times 1}$, $\boldsymbol{C} \in \mathbb{R}^{N \times 1}$ and $\boldsymbol{D} \in \mathbb{R}^{N \times 1}$ denote the projection parameters. For deep sequential modeling, $\boldsymbol{A}$, $\boldsymbol{B}$, $\boldsymbol{C}$ and $\boldsymbol{D}$ are parameters that can be learned by gradient descent. The parameter $\boldsymbol{D}$ is omitted for exposition (*i.e.*, $\boldsymbol{D} = 0$) as $\boldsymbol{D}x(t)$ can be regarded as a skip connection and is easy to compute [19, 20].

**Discretization.** The structured state space [20] and Mamba [17] discretize the above continuous system so as to be tailored for deep representation learning. There are usually two ways for discretization, namely, linear recurrence and discrete convolution. For linear recurrence, instead of a continuous function $x(t)$, a discrete sequence $(x_0, x_1, \cdots)$ is taken as input. Conceptually, we have $x_k = x(k\Delta)$. The state matrix $\boldsymbol{A}$ is approximated as $\overline{\boldsymbol{A}}$ by the zero-order hold rule. The discrete SSM is a sequence-to-sequence map $x_k \mapsto y_k$, given by

$$
\begin{aligned}
u_k &= \overline{\boldsymbol{A}} u_{k-1} + \overline{\boldsymbol{B}} x_k, & \overline{\boldsymbol{A}} &= e^{\Delta \boldsymbol{A}}, \\
y_k &= \overline{\boldsymbol{C}} u_k, & \overline{\boldsymbol{B}} &= \Delta \boldsymbol{B}, \quad \overline{\boldsymbol{C}} = \boldsymbol{C}.
\end{aligned}
\tag{2}
$$

**Selective Scan Mechanism.** Prior SSM methods usually focus on the linear time-invariant scenario. Instead, the selective scan mechanism [17], which is the core of SSM operator in Mamba, learns the dynamism of weights from the input and is more aware of the context information.

The Mamba model is a suitable structure that aligns with our needs. When encoding the image patches as sequential data, once important lesion information is discovered, it can be stored in hidden states and propagated to subsequent sequences. Specifically, after sliding the image $\boldsymbol{x} \in \mathbb{R}^{H \times W \times 3}$ into a variety of patches, the input is formed as a sequences of 2-D patches, each of which has a spatial position of $H/4 \times W/4$. Then, in each Samba block, the bi-directional state space modeling module has both feedforward and backforward SSM, where the selective scan mechanism allows to handle the patches in a recurrent manner. The input patches are traversed along two different scanning paths (horizontal and vertical), and each sequence is independently processed by the SSM. Subsequently, the results are merged to construct a 2D feature map as the final output.

By a bidirectional design, the severity information can be transported to each patch. The local-to-global transportation of severity information plays a vital role in the whole process, especially in the selective mechanism. With the guidance of global severity awareness, the update of hidden states can selectively ignore information about low-level lesions, primarily preserving information about the most severe lesions. Specifically, to encode the 2D images, we follow the design of vision Mamba [69] which processes the input features in the forward and backward directions. As illustrated in Fig. 2, the outputs are gated and added together, while there is a skip connection before input to the EM-based state recalibration module.

### 3.3 EM-based State Recalibration

Another core issue in cross-domain disease grading is the domain generalization ability of the model. Both the intermediate features and the input-dependent parameters in Mamba are affected by the cross-domain variance. To reduce the impact of domain shift, we aim to map the features into a more compact space by feature recalibration. Specifically, the feature distribution of background and grading-related lesions is modeled by a Gaussian Mixture Model (GMM) [42], given by

$$
p(\boldsymbol{f}_n) = \sum_{k=1}^{K} z_{nk} \mathcal{N}(\boldsymbol{f}_n | \boldsymbol{\mu}_k, \boldsymbol{\Sigma}_k),
\tag{3}
$$

where $K$ is the total number of the Gaussian models, $\boldsymbol{f}_n$ is the feature embedding of the $n$-th patch in image $\boldsymbol{x}$, $\boldsymbol{\mu}_k$ and $\boldsymbol{\Sigma}_k$ denote the mean and covariance of the $k$-th Gaussian basis, respectively. For simplicity, we set $\boldsymbol{\Sigma}_k$ as the identity matrix $\boldsymbol{I}$. For easy computation, the mixing coefficients of GMM are left out and the exponential inner dot kernel is used. After that, each Gaussian basis is represented by $\boldsymbol{\mu}_k$, which is called severity base in this paper. These bases are learnable parameters to capture the representation of lesions. In the recalibration process, instead of directly reconstructing the features based on the bases, we estimate the lesion distribution of each image which is initialized by the severity bases. This is to prevent the loss of useful information during the compression. Concretely, we adopt the EM algorithm to estimate the GMM of each image.

Within each iteration, we first estimate $z_{nk}$ in the E-step, which denotes the responsibility of the $k$-th basis to $\boldsymbol{f}_n$. Here we have $1 \leq n \leq N$ and $1 \leq k \leq K$. The posterior probability of $\boldsymbol{f}_n$ given $\boldsymbol{\mu}_k$ can be formulated as $p(\boldsymbol{f}_n | \boldsymbol{\mu}_k) = \mathcal{K}(\boldsymbol{f}_n, \boldsymbol{\mu}_k)$ by a kernel function $\mathcal{K}$. Consequently, estimating the responsibility can be re-formulated into a more general form, given by

$$
z_{nk} = \frac{\mathcal{K}(\boldsymbol{f}_n, \boldsymbol{\mu}_k)}{\sum_{i=1}^{K} \mathcal{K}(\boldsymbol{f}_n, \boldsymbol{\mu}_i)},
\tag{4}
$$

where for simplicity we directly use the exponential inner dot $\exp(\boldsymbol{f}^{\mathrm{T}}\boldsymbol{\mu})$ as the kernel function.

Given the estimated $\mathbf{Z}^t$, the severity base likelihood maximization, functioning as the M-step, is realized by updating $\boldsymbol{\mu}$. As the bases are supposed to be aligned to the embedding space of each image, the weighted sum is used to update the bases, given by

$$\boldsymbol{\mu}_k^{t+1} = \frac{1}{\sum_{m=1}^{N_p} z_{mk}^t} \sum_{n=1}^{N_p} z_{nk}^t \boldsymbol{f}_n, \tag{5}$$

where $t$ refers to the $t$-th iteration and $N_p$ denotes the number of patch embeddings.

Assume that the E-step and M-step execute alternately for $T_c$ iterations and the convergence criterion has been reached [14]. The final $\boldsymbol{\mu}^{T_c}$ and $\mathbf{Z}^{T_c}$ are used to recalibrate the image feature $\boldsymbol{F}$, resulting in $\tilde{\boldsymbol{F}}$. Here $\boldsymbol{\mu}^{T_c} = \{\boldsymbol{\mu}_k^{T_c}\}$ and $\mathbf{Z}^{T_c} = \{z_{nk}^{T_c}\}$ refer to the Gaussian basis and the responsibilities of all the patch embeddings from a sample, respectively. This process is mathematically computed as

$$\tilde{\mathbf{F}} = \mathbf{Z}^{T_c} \boldsymbol{\mu}^{T_c}. \tag{6}$$

Then, the recalibrated feature $\tilde{\mathbf{F}}$ is fed into the next Samba module. During this process, grading-related features are mapped to a more compact space, while style differences introduced by image sources are partially removed. Consequently, the critical information transportation within the Mamba model can be more stable in unseen target domains.

To alleviate the potential unstable issue, moving averaging is adapted to update the bases $\boldsymbol{\mu}^0$ during the training process. After the $T$-th iteration, the generated $\boldsymbol{\mu}^T$ is first averaged over a mini-batch to get $\overline{\boldsymbol{\mu}}^T$. Then, the update of $\boldsymbol{\mu}^0$ with momentum $\alpha \in [0, 1]$ is given by

$$\boldsymbol{\mu}^0 \longleftarrow \alpha\boldsymbol{\mu}^0 + (1-\alpha)\overline{\boldsymbol{\mu}}^T. \tag{7}$$

### 3.4 Theoretical Analysis

Consider the source domain $\mathcal{D}_1 \sim P(\tilde{\mathbf{X}}^{(1)})$ and a certain unseen target domain $\mathcal{D}_k \sim P(\tilde{\mathbf{X}}^{(k)})$, where $k = 2, \cdots, K$. Given a hypothesis $h \in \mathcal{H}$, according to the domain adaptation/generalization bound theory [5, 2], the relation between the target risk $\epsilon_{\mathcal{D}_k}(h)$ and the source risk $\epsilon_{\mathcal{D}_1}(h)$ can be modeled by a relation inequality, given by

$$\epsilon_{\mathcal{D}_k}(h) \leq \epsilon_{\mathcal{D}_1}(h) + d_{\mathcal{H}\Delta\mathcal{H}}\left(P(\tilde{\mathbf{X}}^{(1)}), P(\tilde{\mathbf{X}}^{(k)})\right) + \min_{P(\tilde{\mathbf{X}})\in P(\tilde{\mathbf{X}}^{(1)}), P(\tilde{\mathbf{X}}^{(k)})} \mathbb{E}\left[|h_{\mathcal{D}_1}(x) - h_{\mathcal{D}_k}(x)|\right], \tag{8}$$

where $d_{\mathcal{H}\Delta\mathcal{H}}\left(P(\tilde{\mathbf{X}}^{(1)}), P(\tilde{\mathbf{X}}^{(k)})\right)$ denotes the distribution gap between the source domain and an unseen target domain, and the right-most term refers to minimal total risk over both domains. In other words, the risk of the proposed Samba on the target domain is bounded by the source domain.

## 4 Experiments

### 4.1 Datasets & Evaluation Protocols

**Cross-domain Fatigue Fracture Grading Benchmark** [31] consists of a total number of 1,785 normal X-ray images and 940 X-ray images with fatigue fracture. They are collected from two hospitals with different types of sensors, which we denote as Domain-1 and Domain-2, respectively. These fatigue fracture images were graded into four stages by three physicians according to the severity level. For simplicity, we denote the grades (including the normal grade) from level-1 to level-5.

**Cross-domain Breast Cancer Grading Benchmark** consists of a total of 3644 H&E stained breast invasive ductal carcinoma pathological images from two domains.[2] The first domain contains 2,486 images under the $20\times$ magnification (denoted as Domain-1). The second domain contains 1,158 images under the $40\times$ magnification (denoted as Domain-2). Different magnifications make the image appearance dramatically different. For each experiment setting, one is used as the source

---

[2]https://github.com/YANRUI121/Breast-cancer-grading

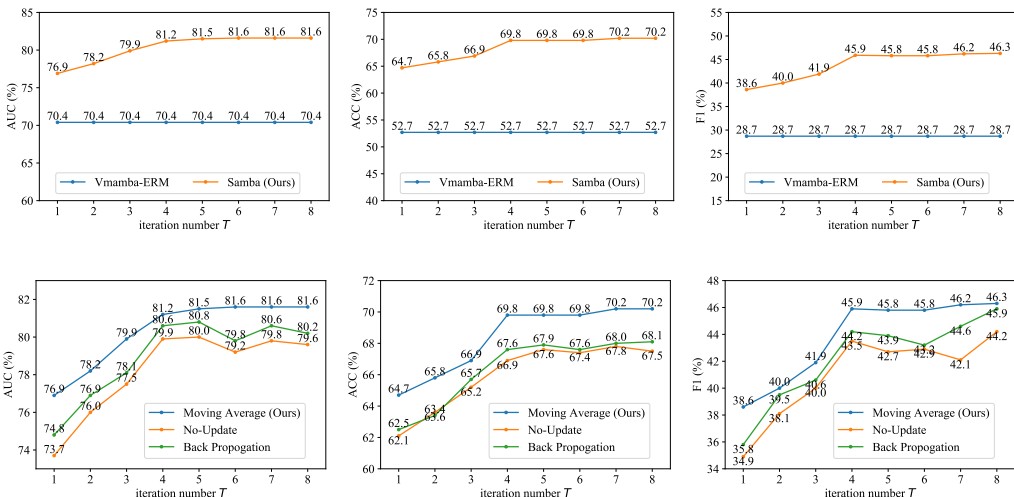

Figure 3: Impact of iteration number $T$ (top row) and severity base (bottom row) updating approaches on generalized medical grading performance. Evaluation metrics AUC, ACC and F1 are presented in percentage (%), from the first to the third column. Domain-1 and Domain-2 in the Fatigue Fracture Grading Benchmark are used as the source and unseen target domain, respectively.

domain and the other is used as the unseen target domain. According to the severity of breast invasive ductal carcinoma, three grades, namely, rare, frequent and abundant, are annotated. For simplicity, we denote them from level-1 to level-3, respectively.

**Cross-domain Diabetic Retinopathy Grading Benchmark** consists of six DR retinal image datasets, namely, DeepDR [33], Messidor [1], IDRID [40], APTOS [3], FGADR [67], and RLDR [49]. Following recent work [11], the single-domain generalization protocol is adapted. Specifically, one of the above six datasets is used as the source domain, and all the rest datasets are used as unseen target domains. Following [11], two extra large-scale datasets, DDR [29] and EyePACS [15] are used to enrich the source domain for each experiment setting. The development of DR is graded into five levels according to the severity, namely, normal, mild nonproliferative diabetic retinopathy (npdr), moderate npdr, severe npdr and pdr. For simplicity, we denote them from level-1 to level-5.

Under the single-domain generalization protocol, three most common evaluation metrics for grading are used, namely, area under the curve (AUC), accuracy (ACC), and F1-score (F1).

## 4.2 Results on Fatigue Fracture Grading Benchmark

We conduct extensive ablation experiments to study the impact of iteration number $T$ and optimization of severity basis $\boldsymbol{\mu}$ on the unseen target domain. Images from the first clinical center (Domain-1) are used as the source domain, while images from the second clinical center (Domain-2) are used as the unseen target domain for testing. The vanilla Mamba [35] under the Empirical Risk Minimization (ERM) is the baseline.

**Iteration Number $T$.** In the EM algorithm, the iteration number $T$ plays an important role, because it implements the approximation by iteratively conduct the E and M step. Keeping other hyper-parameters and module designs the same, we report the results when the iteration number $T$ of the EM algorithm varies from 1 to 8. The top row of Fig. 3 shows how $T$ impacts the AUC, ACC and F1-score on the unseen target domain. Notice that, the VMamba-ERM baseline does not have EM-based state recalibration. Therefore, the performance of Vmamba-ERM is consistent to $T$. A too-small $T$ does not reach the convergence criterion, and reduces the effectiveness of feature recalibration. Therefore, a clear performance decline on all the metrics is observed. On the other hand, when $T$ is too large, the representation ability saturates, resulting in little performance improvement, while wasting computation resources.

**Severity Base Update.** We further study how different optimization approaches of the severity base $\boldsymbol{\mu}_k$ impact the generalization performance on unseen target domain. We study three different settings, namely, no update, only back propagation, and moving average (Eq. 7). The bottom row of Fig. 3

Table 1: Effectiveness of the proposed Samba on recurrent patch modeling. Domain-1 and Domain-2 in the Fatigue Fracture Grading Benchmark are used as the source and unseen target domain, respectively. Metrics presented in percentage (%).

| Method | ACC ↑ | AUC ↑ | F1 ↑ |
|---|---|---|---|
| LSTM [22] | 39.8 | 50.2 | 18.6 |
| UR-LSTM [18] | 43.3 | 61.8 | 20.9 |
| UR-GRU [18] | 45.7 | 65.1 | 22.4 |
| ViT [48] | 50.0 | 69.3 | 26.5 |
| VMamba [69] | 52.7 | 70.4 | 28.7 |
| Samba | **76.2** | **81.5** | **45.8** |

Table 2: Ablation study on each component. BSSM: Bi-directional State Space Modeling; ESR: EM-based State Recalibration. Experiments on the Fatigue Fracture Grading Benchmark. Domain-1 (×20)/Domain-2 (×40) is used as source/target domain. Metrics in percentage (%).

| Components | | | Evaluation Metric | | |
|---|---|---|---|---|---|
| VMamba | BSSM | ESR | ACC | AUC | F1 |
| ✓ | ✗ | ✗ | 52.7 | 70.4 | 28.7 |
| ✓ | ✓ | ✗ | 57.9 | 72.1 | 33.6 |
| ✓ | ✓ | ✓ | **76.2** | **81.5** | **45.8** |

Table 3: Category-wise performance and computational cost comparison between VMamba-ERM and the proposed Samba. Experiments are conducted on the DG Breast Cancer Grading Benchmark. Domain-1 (×20)/Domain-2 (×40) is used as source/target domain. Metrics in percentage (%).

| Method | Backbone | Computation | | Domain-1 as Source | | | |
|---|---|---|---|---|---|---|---|
| | | GFLOPs | Para. | level-1 | level-2 | level-3 | avg. |
| ERM | VMama-T | 3.7 | 32.7 | 22.1 | 51.5 | 36.1 | 40.4 |
| Samba | | 5.5 | 32.7 | **40.5** | **70.7** | **42.0** | **54.8** |
| ERM | VMama-S | 7.9 | 63.4 | 26.7 | 60.6 | 38.1 | 50.1 |
| Samba | | 11.3 | 63.4 | **47.1** | **71.5** | **43.7** | **56.1** |
| ERM | VMama-B | 14.0 | 112.4 | 27.8 | 75.4 | 38.2 | 54.9 |
| Samba | | 19.6 | 112.4 | **44.8** | **82.5** | **45.2** | **60.5** |

Table 4: Impact of the number of components $K$ in GMM. Experiments are conducted on the DG Breast Cancer Grading Benchmark. Domain-1 (×20)/Domain-2 (×40) is used as source/target domain. Metrics presented in percentage (%).

| $K$ value | ACC ↑ | AUC ↑ | F1 ↑ |
|---|---|---|---|
| 16 | 58.6 | 70.0 | 56.0 |
| 32 | 59.2 | 71.1 | 57.2 |
| 48 | 60.4 | 72.0 | 58.9 |
| 64 | **60.5** | **72.3** | **59.1** |
| 96 | 60.4 | 72.2 | 58.8 |
| 128 | 59.5 | 71.0 | 57.9 |

shows the results of the above three settings under a variety of iteration number $T$. Using moving average to update the severity base $\boldsymbol{\mu}_k$ is able to improve the performance substantially. It may be explained that the proposed state recalibration is differentiable, thereby enabling the application of back-propagation to update $\boldsymbol{\mu}_0$. However, the stability of the update cannot be guaranteed due to the EM iterations. Moving average can update $\boldsymbol{\mu}_0$ to avoid collapse.

**Effectiveness on Recurrent Patch Modeling.** The proposed Samba realizes the recurrent patch modeling by harnessing the selective state space model. To demonstrate its effectiveness compared with other recurrent or long-context based representation learning methods, we compare the proposed Samba with vanilla VMamba [69], Vision Transformer [48], LSTM [22] and UR-LSTM [18]. Table 1 shows that the proposed Samba has a stronger generalization performance on the unseen target domain, noticeably outperforming the second-best by 23.5%, 11.1% and 17.1% in terms of accuracy, AUC and F1-score, respectively.

**Ablation Studies on Each Component.** On top of the VMamba baseline, two key components, namely, Bi-directional State Space Modeling (BSSM) and EM-based State Recalibration (ESR), are evaluated. The experiments are conducted on the DG Fatigue Fracture Grading Benchmark. Domain-1/Domain-2 is used as the source/target domain, respectively. The results are reported in Table 2. It is observed that BSSM contributes to an ACC, AUC and F1 improvement of 5.2%, 1.7% and 4.9%, respectively. ESR contributes to an ACC, AUC and F1 improvement of 18.3%, 9.4% and 12.2%, respectively.

## 4.3 Results on Breast Cancer Grading Benchmark

**Grade-wise Improvement Analysis.** We provide a break-down analysis on the grade-wise performance of the proposed Samba and the baseline, *i.e.*, VMamba under the empirical risk minimization (ERM). Table 3 reports the performance. The proposed Samba shows a significant performance improvement on each grade level. Especially, on level-1, level-2 and level-3, the accuracy improvement over the ERM baseline is 17.0%, 7.1% and 7.0%, respectively. Compared to VMamba-ERM baseline, the EM-based State Recalibration in Samba models the feature distribution of lesions via Gaussian Mixture Models with learnable severity bases, and re-estimates by E-M algorithm. The grading features are mapped to a more compact space, and are more stable on unseen target domains.

**Ablation Studies on the number of Gaussian components $K$.** We study how the number of components $K$ impacts the generalization ability on the unseen target domain. By default $K$ is set

to 64, and we further test the performance when $K$ is set to 16, 32, 48 and 96, respectively. The results are reported in Table 4. When $K$ is set to 64, the proposed Samba achieves the best grading performance, *i.e.*, 60.5%, 72.3% and 59.1% in terms of ACC, AUC and F1-Score, respectively.

**Scalability to Clinical Computational Pathology.** The domain shift in the Cross-domain Breast Cancer Grading Benchmark is from a machine learning perspective, and only handles the magnification difference. However, from a clinical perspective, the computational pathology has to handle the domain shift from not only the magnification difference, but also the staining procedure. However, most existing clinical computational pathology datasets only support the classification task, *i.e.*, separating *tumor* category from *normal* category, which is not strongly relevant to our grading task. Appendix A.4 studies the performance of the proposed Samba along with the vanilla VMamba baseline on the CAMELYON17 dataset [3] for a clinical sanity check.

## 4.4 Results on Diabetic Retinopathy Grading Benchmark

**Comparison with State-of-the-art.** We compare the proposed Samba with methods from three primary categories: 1) generic domain generalization methods, including Mixup [61], MixStyle [64], DDAIG [63], ATS [55], Fishr [41], and MDLT [56]; 2) state-of-the-art DR grading methods, which focus on DR grading without explicitly addressing domain generalization, including GREEN [34], CABNet [21], Swin-Transformer [36] and MIL-ViT [7]; 3) domain-generalized DR grading methods, including DRGen [4] and GDRNet [11]. Additionally, the vanilla Mamba [35] results under Empirical Risk Minimization (ERM) are provided as a baseline reference. By default, the results are cited directly from [11].

Table 5 presents a comparison between Samba and existing methods within the context of single-domain generalization. Notably, for DG grading tasks, the metric of accuracy and F1-score are more meaningful than AUC, as the AUC can be made artificially high due to the large amount of negative samples belonging to other stages. Therefore, we only involve ACC and F1 for comparison.

The proposed Samba achieves a substantial improvement over state-of-the-art domain generalized DR grading methods. Especially, on APTOS, DeepDR, FGADR, Messidor and RLDR, it outperforms the second-best in terms of ACC and F1 by 5.2% and 2.2%, 27.2% and 5.7%, 60.7% and 31.3%, 6.7% and 1.3%, 28.3% and 4.7%, respectively. The significant improvement on FGDAR may be explained that it has a different severity-level sample distribution than other datasets. The samples without DR (level-1) only occupy only 5.5% among all the training samples, which are far less than others (*e.g.*, level-1 samples occupy 49.3% in APTOS). Therefore, existing methods may overfit other severity levels and underfit level-1. In contrast, the selective scan mechanism of the Vmamba-ERM and Samba is robust to this severity distribution shift. The EM state re-calibration in Samba makes the feature space more compact, and improves the generalization.

Additionally, Samba shows a marked improvement over the baseline VMamba model under Empirical Risk Minimization. Especially, on APTOS, DeepDR, FGADR, IDRID, Messidor and RLDR, it outperforms the VMamba-ERM baseline in terms of ACC and F1 by 1.3% and 1.7%, 2.2% and 2.1%, 3.0% and 1.6%, 3.7% and 2.6%, 7.3% and 2.7%, 7.4% and 3.4%, respectively. These results demonstrate its effectiveness in handling domain gap.

**Understanding Recurrent Patch Modeling.** We validate if the proposed recurrent patch modeling can store and transport the lesion information. An intuitive way is to inspect the correlation between the patch embeddings before and after the recurrent patch modeling. Therefore, we extract the patch embeddings before and after the fourth block. We compute the correlation matrix between the patch-wise embeddings and visualize the results in Fig. 4. After processed by the Recurrent Patch Modeling module, more regions in the correlation matrix have higher responses. Specifically, after passing through certain high-response positions, the relevant information is transmitted to the subsequent patches in the forward direction. The high-response patches have grade-related lesions and the information is transported in the recurrent process.

**t-SNE Visualization.** To assess the generalization capacity of the proposed Samba, we analyze the feature distribution across the source and unseen target domains using t-SNE visualization in Fig. 5, which compares the t-SNE plots of the ERM baseline (left) and the proposed Samba (right). The results indicate that, after the EM-based state feature calibration, the proposed Samba enables

---

[3] https://camelyon17.grand-challenge.org/

Table 5: Performance comparison of the proposed Samba and existing domain generalized DR grading methods under the single-domain generalization protocol. Evaluation metrics include ACC and F1 (in percentage %). Top three results are highlighted as best , second and third , respectively.

| Method | APTOS | | DeepDR | | FGADR | | IDRID | | Messidor | | RLDR | | Average | |
|---|---|---|---|---|---|---|---|---|---|---|---|---|---|---|
| | ACC↑ | F1↑ | ACC↑ | F1↑ | ACC↑ | F1↑ | ACC↑ | F1↑ | ACC↑ | F1↑ | ACC↑ | F1↑ | ACC↑ | F1↑ |
| *ResNet-50 based:* | | | | | | | | | | | | | | |
| Mixup [61] | 49.4 | 30.2 | 49.7 | 33.3 | 5.8 | 7.4 | 64.0 | 32.6 | 63.0 | 32.6 | 27.7 | 27.0 | 43.3 | 27.2 |
| MixStyle [64] | 48.8 | 25.0 | 32.0 | 14.6 | 7.0 | 7.9 | 53.5 | 19.4 | 57.6 | 16.8 | 18.3 | 6.4 | 36.2 | 15.0 |
| GREEN [34] | 52.6 | 33.3 | 44.6 | 31.1 | 5.7 | 6.9 | 60.7 | 33.0 | 54.5 | 33.1 | 31.9 | 27.8 | 41.7 | 27.5 |
| CABNet [21] | 52.2 | 30.8 | 55.4 | 32.0 | 6.1 | 7.5 | 62.7 | 31.7 | 63.8 | 35.3 | 23.0 | 25.4 | 43.8 | 27.2 |
| DDAIG [63] | 48.7 | 31.6 | 38.5 | 29.7 | 5.0 | 5.5 | 60.2 | 33.4 | 69.1 | 35.6 | 25.4 | 23.5 | 41.2 | 26.7 |
| ATS [55] | 51.7 | 32.4 | 52.4 | 33.5 | 5.3 | 5.7 | 66.6 | 30.6 | 64.8 | 32.4 | 24.2 | 23.9 | 44.2 | 26.4 |
| Fishr [41] | 61.7 | 31.0 | 61.0 | 30.1 | 6.0 | 7.2 | 48.0 | 30.6 | 52.0 | 33.8 | 19.3 | 21.3 | 41.3 | 25.7 |
| MDLT [56] | 53.3 | 32.4 | 50.2 | 33.7 | 7.1 | 7.8 | 61.7 | 32.4 | 58.9 | 34.1 | 29.0 | 30.0 | 43.4 | 28.4 |
| DRGen [4] | 60.7 | 35.7 | 39.4 | 31.6 | 6.8 | 8.4 | 67.7 | 30.6 | 64.5 | 37.4 | 19.0 | 21.2 | 43.0 | 27.5 |
| GDRNet [11] | 52.8 | 35.2 | 40.0 | 35.0 | 7.5 | 9.2 | 70.0 | 35.1 | 65.7 | 40.5 | 44.3 | 37.9 | 46.7 | 32.2 |
| *ViT based:* | | | | | | | | | | | | | | |
| MIL-ViT [7] | 61.8 | 36.8 | 38.2 | 36.3 | 8.7 | 9.3 | 68.6 | 31.1 | 67.7 | 40.7 | 28.1 | 34.5 | 45.5 | 31.5 |
| Swin-T [36] | 64.0 | 36.7 | 31.0 | 32.7 | 6.0 | 7.8 | 70.4 | 38.1 | 65.6 | 39.8 | 27.5 | 34.5 | 44.1 | 31.6 |
| *VMamba based:* | | | | | | | | | | | | | | |
| ERM | 64.6 | 36.2 | 65.0 | 38.6 | 65.2 | 38.9 | 65.2 | 39.1 | 65.1 | 39.1 | 65.2 | 39.2 | 65.1 | 38.5 |
| Samba (Ours) | 65.9 | 37.9 | 67.2 | 40.7 | 68.2 | 40.5 | 68.9 | 41.7 | 72.4 | 41.8 | 72.6 | 42.6 | 69.2 | 40.9 |

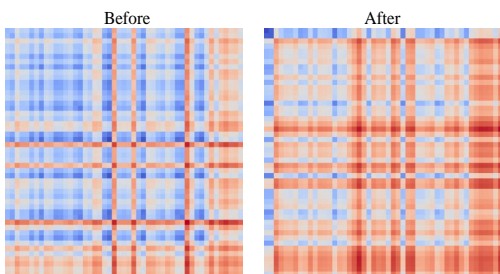

Figure 4: The correlation matrix of each patch embedding before and after processed by the recurrent patch modeling in the forward direction, denoted as 'Before' and 'After' respectively. The higher correlation, the more red a cell is.

Figure 5: T-SNE visualization of the feature space from the ERM baseline (left), and the proposed Samba (right). APTOS is chosen as the source domain and the rest datasets are used for as target domains.

feature embeddings from different domains to achieve a more uniform distribution, thereby reducing the domain gap. This improved uniformity in the feature space suggests that Samba can enhance generalization, contributing to better performance on unseen domains.

## 5 Conclusion

In this paper, we aimed to tackle a practical but challenging task, learning domain generalized medical image grading. We mainly focused on two issues: the identification of decisive lesions and the impact caused by inter-domain differences. The proposed severity-aware recurrent modeling adopts a state space model to store and transport the severity information from local to global. To further mitigate the impact of cross-domain variants, an EM-based state recalibration was designed to map the patch embeddings into a compact space. The proposed method can be used in a variety of disease grading scenarios, providing an effective tool for automatic medical image analysis.

**Limitation Discussion & Broader Societal Impact.** The feature distribution of lesions is modeled by the Gaussian mixture model and estimated by the Expectation-Maximization algorithm. However, when the training source domain has severe class imbalance, the estimated probability distribution by the proposed Samba may not necessarily reflect the domain-agnostic lesion distribution. Nevertheless, the proposed method can be combined with other techniques specifically designed for addressing class imbalance. The proposed method advances the versatility of automatic disease diagnosis, which benefits the human well beings. We do not envision negative societal impact.

**Acknowledgments and Disclosure of Funding.** This work was supported by the Science and Technology Major Project of Guangxi (AA22096030 and AA22096032), and National Key R&D Program of China under Grant (2020AAA0109500 and 2020AAA0109501).

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

# A  Appendix / supplemental material

## A.1  Severity Base Normalization

During the iteration steps $t = 1, \cdots, T$, the severity base $\boldsymbol{\mu}_k^{(1),t}$ may not deviate too much from each other, which otherwise can lead to collapse when back propagation. We study the scenarios when no normalization, $L$-1 normalization and $L$-2 normalization are used on these severity basis. Fig. 6 shows the results of the above three settings when under a variety of iteration number $T$ on the unseen target domain. $L$-2 normalization achieves the best performance on all the metrics, especially when $T$ becomes larger.

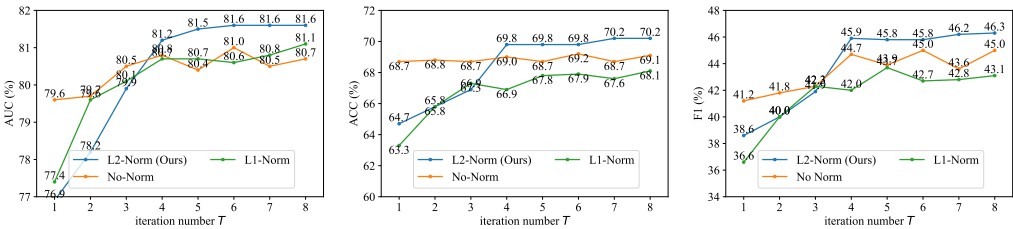

Figure 6: Impact of severity base normalization on generalized medical grading performance. Evaluation metrics AUC, ACC and F1 are presented in percentage (%). Domain-1 and Domain-2 in the Fatigue Fracture Grading Benchmark are used as the source and unseen target domain, respectively. Better zoom in to view.

## A.2  Understanding Recurrent Patch Modeling

Fig. 4 in the main text only visualizes the correlation between the patch embeddings before and after the recurrent patch modeling when trained on the APTOS dataset. Here we further demonstrate the results from FGADR, IDRID, Messidor and RLDR. They are visualized in Fig. 7a, b, c and d, respectively. On all these datasets, we can observe a common pattern. After processed by the Recurrent Patch Modeling module, more cells in the correlation matrix have higher response. Usually, a handful of the patches inside the image have grade-related lesions. After the processing of our module, the information of these grade-related lesions is transported to other patches. It allows the model to perceive a more global-wise representation. Consequently, more patches that contain the grade-related lesion information are activated, and more cells are highly responded in the correlation matrix.

## A.3  Visualize and Understand the Severity Level

We model the relation between patch embedding from SSM and severity level by drawing inspiration from the class activation map (CAM) mechanism [62]. Specifically, we take the patch embeddings from the last Samba block as input to generate the per-level severity activation patterns. Then, the activated severity patterns are displayed on the original images. We use FGADR as the unseen target domain. The results are shown in Fig. 8, where the activated patches are highlighted in blue boxes. From the first to the fifth row, the samples from level-1 to level-5 are provided accordingly. From the first to the fifth column, the patch activation maps from level-1 to level-5 are displayed. Notice that, as level-1 refers to the normal scenario, each sample has activations on level-1, meaning some patches are normal.

## A.4  Application to Computational Pathology Classification

It is important to note that the domain shift in the Breast Cancer Grading Benchmark is technically from a machine learning perspective, and only handles the domain shift from the magnification difference. However, from a clinical perspective, the computational pathology has to handle the domain shift not only from the magnification difference, but also from the staining procedure. Therefore, it is beneficial to test if the hypothesis works in a real-world computational pathology scenario. However, a bottleneck is that, most clinical computational pathology dataset so far conducts the classification task, *i.e.*, separating *tumor* category from *normal* category. Therefore, in this

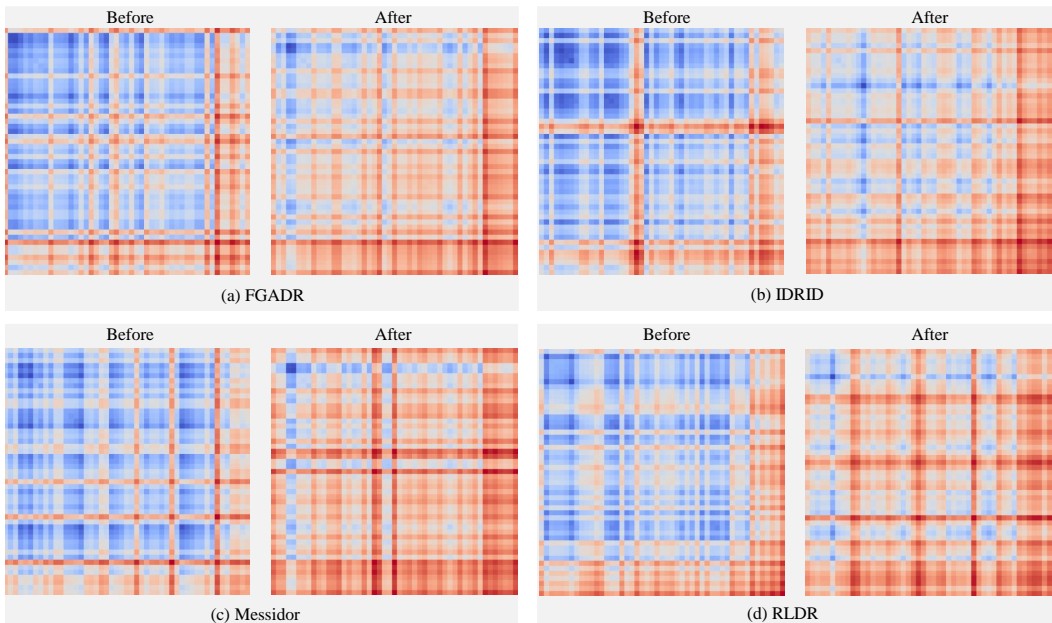

Figure 7: The correlation matrix of each patch embedding before and after processed by the recurrent patch modeling in the forward direction, denoted as 'Before' and 'After', respectively. The higher correlation, the more red a cell is.

Table 6: Impact of the number of components $K$ in GMM on tumor classification performance from unseen target domains. Experiments are conducted on the CAMELYON17. Domain-1 is used as source domain. The rest four are used as unseen target domains. Metrics presented in percentage (%).

| $K$ value | Domain-2 | | | Domain-3 | | | Domain-4 | | | Domain-5 | | |
|---|---|---|---|---|---|---|---|---|---|---|---|---|
| | ACC | AUC | F1 | ACC | AUC | F1 | ACC | AUC | F1 | ACC | AUC | F1 |
| 16 | 81.80 | 92.09 | 79.96 | 79.45 | 88.96 | 76.60 | 83.92 | 95.38 | 82.10 | 75.84 | 81.99 | 74.16 |
| 32 | 82.68 | 93.81 | 80.95 | 80.39 | 91.28 | 77.97 | 84.38 | 96.20 | 82.96 | 76.53 | 82.46 | 74.92 |
| 48 | 84.06 | 94.25 | 81.83 | 81.84 | 92.60 | 78.49 | 85.87 | 97.16 | 83.87 | 77.69 | 83.85 | 75.04 |
| 64 | **84.59** | **95.67** | **83.10** | **82.48** | **93.32** | **79.85** | **86.50** | **97.82** | **85.13** | **78.64** | **84.70** | **75.32** |
| 96 | 84.27 | 95.48 | 82.53 | 82.06 | 92.90 | 79.44 | 86.16 | 97.45 | 84.80 | 78.15 | 84.28 | 74.86 |
| 128 | 83.65 | 94.70 | 81.97 | 81.50 | 92.41 | 78.36 | 85.47 | 96.90 | 84.62 | 77.38 | 83.66 | 74.14 |

subsection, the proposed Samba along with the vanilla VMamba baseline are benchmarked on the CAMELYON17 dataset [4] for the cross-domain computational pathology classification task.

The first experiment is the impact of the number of components $K$ in GMM, where ACC, AUC and F1 are used as evaluation metric. The results are reported in Table 6. By default $K$ is set to 64, and we further test the performance when $K$ is set to 16, 32, 48 and 96, respectively. When $K$ is set to 64, the proposed Samba achieves the best grading performance. This observation is consistent to the performance on Cross-domain Breast Cancer Grading Benchmark, where a number of 64 Gaussians achieves the optimal performance.

The second experiment is to analyze the trade off between the classification performance and the baseline model. Both the VMamba-ERM and the proposed Samba are tested, where only accuracy is used as the evaluation metric. The results are reported in Table 7. The trend is the same as the trend on the Breast Cancer Grading Benchmark. Using Samba on each type of the VMamba backbone shows a clear performance improvement on unseen domains.

### A.5 Attention Maps on Unseen Domains by Samba

Fig. 9 and Fig. 10 show some attention maps of the Samba on unseen retinal images. The proposed Samba is able to model the recurrent relation among patches. Therefore, the activation regions can general cover the lesions and are more robust to the domain shift.

---

[4]https://camelyon17.grand-challenge.org/

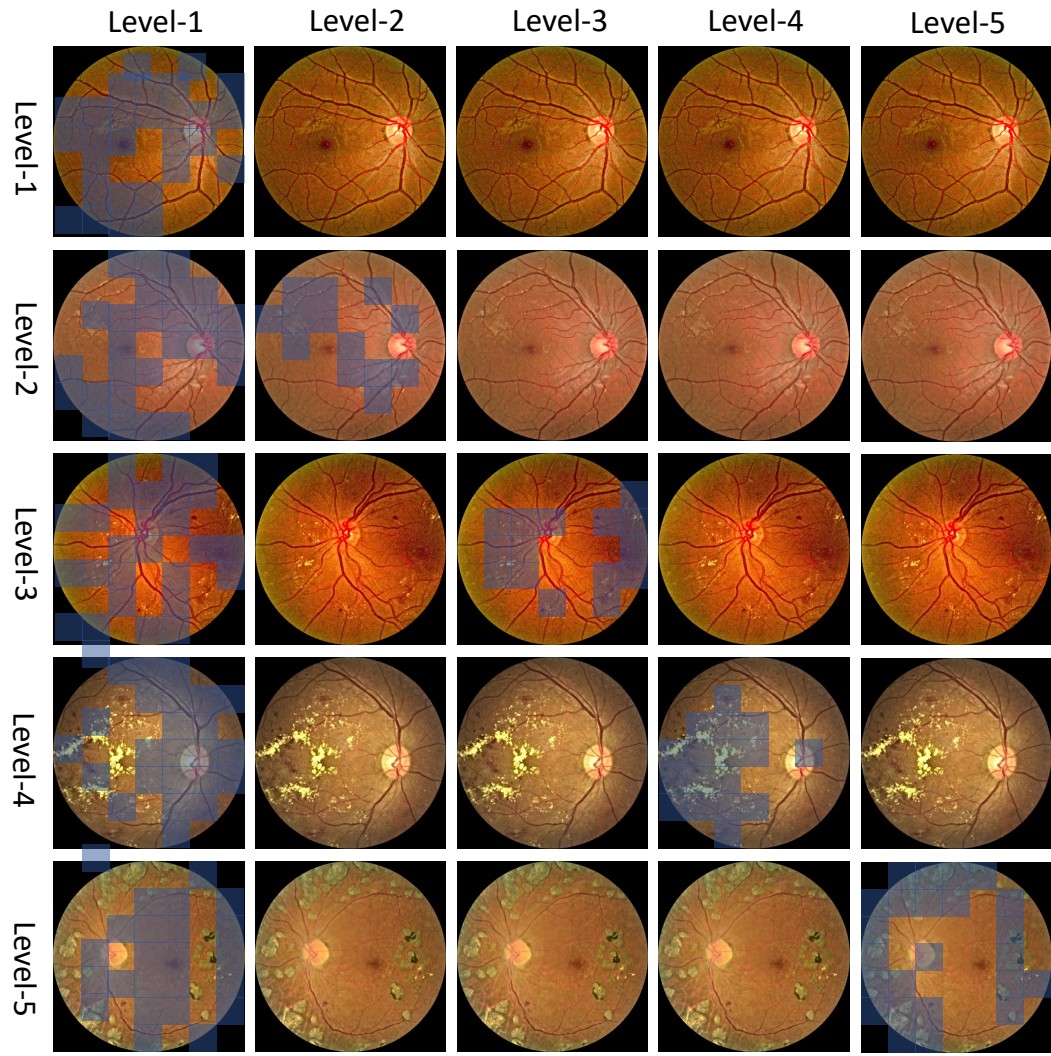

Figure 8: Per-severity activation map of the proposed Samba. From the first to fifth column are the activated patches from level-1 to level-5, highlighted in blue boxes. From the first to the fifth row are the samples with an annotation from level-1 to level-5. FGADR is the unseen target domain.

Table 7: Classification performance comparison between VMamba-ERM and the proposed Samba. Experiments are conducted on the CAMELYON17 dataset for cross-domain tumor classification. Domain-1 is used as the source domain, while the rest four are used as unseen target domains. Metrics in percentage (%).

| Method | Backbone | Domain-1 as Source | | | | |
|--------|----------|----------|----------|----------|----------|------|
| | | Domain-2 | Domain-3 | Domain-4 | Domain-5 | avg. |
| ERM | VMama-T | 70.08 | 67.29 | 72.96 | 63.16 | 68.37 |
| Samba | | **78.74** | **76.15** | **80.06** | **71.05** | **76.50** |
| ERM | VMama-S | 72.86 | 69.50 | 75.08 | 65.72 | 70.79 |
| Samba | | **81.01** | **78.96** | **82.75** | **73.88** | **79.15** |
| ERM | VMama-B | 76.23 | 74.17 | 79.53 | 69.87 | 74.95 |
| Samba | | **84.59** | **82.48** | **86.50** | **78.64** | **83.05** |

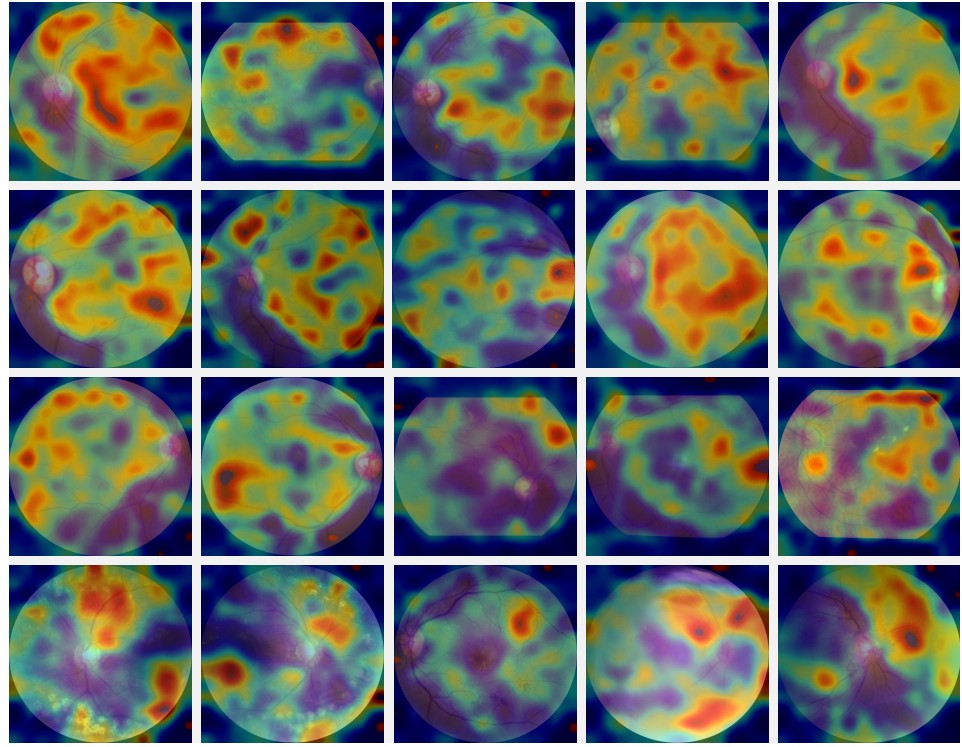

Figure 9: Attention maps of the proposed Samba on retinal images from unseen domains.

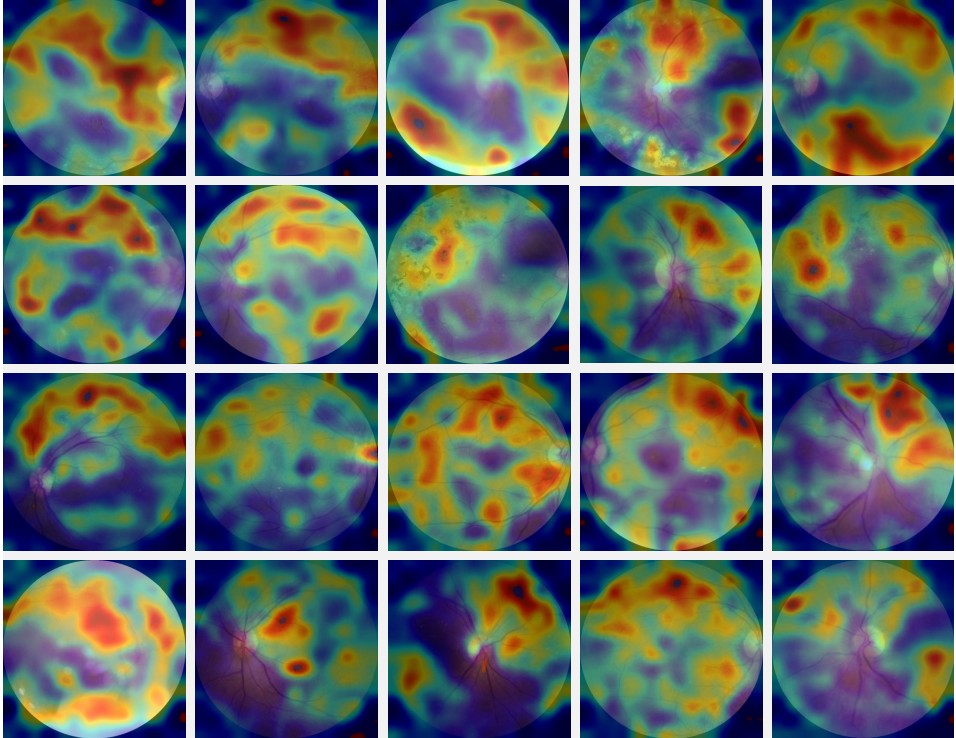

Figure 10: Attention maps of the proposed Samba on retinal images from unseen domains.

