# OpenReview forum: "Samba: Severity-aware Recurrent Modeling for Cross-domain Medical Image Grading"
_NeurIPS.cc/2024/Conference — NeurIPS 2024 poster_

### Official Review · Reviewer_hFq3 · 2024-06-25

**Soundness:** 3
**Presentation:** 3
**Contribution:** 3
**Rating:** 6
**Confidence:** 3

**Summary:**

Accurate disease grading in medical image analysis is challenging due to the variability within disease levels and the similarity between adjacent stages. Additionally, models must handle data from unseen target domains, where differences in feature distribution can significantly reduce performance. To address these issues, this paper proposes the Severity-aware Recurrent Modeling (Samba) method, which encodes image patches sequentially to capture severity information and employs an Expectation-Maximization based recalibration mechanism to handle cross-domain variations. The method also uses a Gaussian Mixture Model to model feature distributions and reconstructs intermediate features using learnable severity bases.

**Strengths:**

- Addresses a significant problem in medical image analysis where different grading can appear differently in the data.
- Different diseases and imaging modalities have been evaluated.
- Use of publicly available datasets.
- Code will be made publicly available upon acceptance.

**Weaknesses:**

- Some parts need more detailed explanation. Although space is limited, certain sections assume a high level of pre-knowledge. For example, more explanation on Samba would be useful.
- For me it is not really clear how the patches are used in a recurrent manner and what their sequence should be. Can you please elaborate more on the recurrent part.
- There is no study demonstrating that the model specifically attends to patches related to severity. Is there a way to illustrate this?

**Questions:**

- Adding specific numbers and metrics to the abstract would be beneficial. Currently, lines 18-19 are quite vague.
- On page 7, the authors likely intend to refer to Figure 3 for the results, rather than Figure 6.
- In Figure 5, the T-SNE plot is difficult to interpret due to small icons. Could the datasets be represented by icons and the severity by color for better qualitative evaluation?

**Limitations:**

The authors addressed the limitations of the work.

---

> ### Author Rebuttal · Authors · 2024-08-05
>
> **Q1**: Some parts need more detailed explanation. Although space is limited, certain sections assume a high level of pre-knowledge. For example, more explanation on Samba would be useful.
>
> **R**: Thanks for your valuable feedback.
> We will provide more explanations of the proposed Samba, such as:
> 1) After proceeded by BSSM, the feature embedding of each patch embedding $\boldsymbol{f}_n$ is firstly projected into the latent space by a linear layer to get the target state embedding $\boldsymbol{x}_n$. The severity base $\boldsymbol{\mu}_n$ is modeled as a mixture of Gaussian from $\boldsymbol{x}_n$.
> Then, the E-M algorithm iteratively approximates $\boldsymbol{\mu}_n$ to $\boldsymbol{x}_n$.
> After convergence, the embedding is fed into the rest module.
> 2) Each Gaussian is initialized by the Kaiming initialization.
> 3) The E-M empirically implements 3 iterations, according to the observation in Fig.3.
> 4) The number of Gaussian kernels $K$ in L171 will be defined when first introducing.
> 5) The specialization of Samba for cross-domain tasks, especially,
> the feature distribution shift between the source domain and unseen target domains affects not only the intermediate features but also the selective scan mechanism of Mamba, which poses a
> clear performance drop of Mamba on unseen target domains. To address issue, the proposed method not only introduces both forward and backward directions to primarily preserve information about the most severe lesions, but also introduces a EM-based State Recalibration mechanism to compact the feature space so that the feature distribution is less sensitive to the domain shift.
>
> **Q2**: It is not really clear how the patches are used in a recurrent manner and what their sequence should be. Can you please elaborate more on the recurrent part?
>
> **R:** Thanks for your valuable feedback.
> After sliding the image $\mathbf{I} \in \mathbb{R}^{H \times W \times 3}$ into a variety of patches, the input is formed as a sequences of 2-D patches, each of which has a spatial position of $H/4 \times W/4$.
> Then, in each Samba block, the Bi-directional State Space Modeling module has both feedforward and backforward SSM, where the selective scan mechanism allows to handle the patches in a recurrent manner.
> Specifically, the 2-D selective scan mechanism in both components is directly inherited from [15].
> The input patches are traversed along two different scanning paths (horizontal and vertical), and each sequence is independently processed by the SSM.
> Subsequently, the results are merged to construct a 2D feature map as the final output.
>
> We will enrich these details accordingly.
>
> **Q3**: There is no study demonstrating that the model specifically attends to patches related to severity. Is there a way to illustrate this?
>
> **R**: Thanks for your valuable suggestion.
> We model the relation between patch embedding from SSM and severity level by drawing inspiration from the class activation map (CAM) mechanism [a].
> We take the patch embeddings from the last Samba block as input, so as to generate the per-level severity activation patterns.
> Then, the activated severity patterns are displayed on the original images.
> We use FGADR as the unseen target domain.
> The results are shown in Fig.~R1, where the activated patches are highlighted in blue boxes.
> From the first to the fifth row, the samples from level-1 to level-5 are provided accordingly.
> From the first to the fifth column, the patch activation map from level-1 to level-5 is generated by the aforementioned methods.
> Notice that, as level-1 refers to the normal scenario, each sample has activations on level-1, meaning some patches are normal.
>
> [a] Zhou, Bolei, et al. "Learning deep features for discriminative localization." Proceedings of the IEEE conference on computer vision and pattern recognition. 2016.
>
> **Q4**: Adding specific numbers and metrics to the abstract would be beneficial. Currently, lines 18-19 are quite vague.
>
> **R**: Thanks for your valuable suggestion.
> We will accordingly add the following in the introduction:
> *Extensive experiments show that the proposed Samba outperforms the VMamba baseline by an average accuracy of 23.5\%, 5.6\% and 4.1\% on the cross-domain grading of fatigue fracture, breast cancer and diabetic retinopathy, respectively.*
>
> **Q5**: On page 7, the authors likely intend to refer to Figure 3 for the results, rather than Figure 6.
>
> **R**: Sorry for the typo. We will correct it from Fig.6 to Fig.3.
>
> **Q6**: In Fig.5, the T-SNE plot is difficult to interpret due to small icons. Could the datasets be represented by icons and the severity by color for better qualitative evaluation?
>
> **R:** Thanks for your valuable suggestion, and we have modified it accordingly.
> Please refer to Fig.~R3 in the attached PDF file for reference.
>
> Finally, should you have further suggestions and comments, we are glad to incorporate during the discussion stage.

---

> > ### Comment · Reviewer_hFq3 · 2024-08-08
> >
> > I can certainly appreciate the significant effort put into the rebuttal and I remain as a weak accept.

---

> > > ### Author Response · Authors · 2024-08-08
> > > **Response to Reviewer hFq3**
> > >
> > > Thanks for your swift response. We are glad to see your questions resolved.
> > >
> > > We will improve our paper carefully according to your valuable suggestions.

---

### Official Review · Reviewer_b4sx · 2024-07-09

**Soundness:** 3
**Presentation:** 3
**Contribution:** 2
**Rating:** 5
**Confidence:** 4

**Summary:**

The authors introduce a new method named Severity-aware Recurrent Modeling (Samba) for disease grading in medical imaging. Specifically, they propose encoding image patches in a recurrent manner to accurately capture decisive lesions and transmit critical information from local to global contexts. Additionally, an Expectation-Maximization (EM) based state recalibration mechanism is designed to map feature embeddings into a compact space, thereby reducing the impacts of cross-domain variations.

**Strengths:**

1.	The proposed severity-aware recurrent modeling uses a state space model to store and transmit severity information from local to global, which is valuable for the classification of medical images with small lesion areas.
2.	For domain adaptation tasks, an EM-based state recalibration mechanism was also proposed and its effectiveness was validated in experiments.
3.	The interpretability visualization of the experimental results is excellent.

**Weaknesses:**

1.	The significance of using the Mamba architecture in cross-domain tasks is uncertain; in fact, restricting the Mamba module may lead to excessive specialization.
2.	The ablation experiments regarding the specific structure of Samba are not sufficiently comprehensive.
3.	The open-source code is not explicitly provided.

**Questions:**

1.	Among the compared methods mentioned in Table 4, some ViT-based and ResNet-50-based methods are not specifically designed for domain adaptation tasks. Is this comparison fair?
2.	The article states: "The Mamba model is a suitable structure that aligns with our needs. Guided by global severity awareness, the update of hidden states can selectively ignore information about low-level lesions, primarily preserving information about the most severe lesions." Can you explain how SSM selects severe lesion information? A theoretical justification should be provided, as this is a point of concern.
3.	In the comparative experiments with other SOTA methods, are all the source domains supplemented with the two additional large-scale datasets DDR [24] and EyePACS [13]?

**Limitations:**

The authors have addressed the limitations of the current work.

---

> ### Author Rebuttal · Authors · 2024-08-05
>
> **Q1**: Significance of using Mamba in cross-domain tasks; restricting Mamba module may lead to excessive specialization.
>
> **R**: Thanks for your valuable comments, so that we could have a chance to clarify the generalization and universality of the proposed Samba. Specially,
>
> (1) The feature distribution shift between the source domain and unseen target domains affects not only the intermediate features but also the selective scan mechanism of Mamba, which poses a clear performance drop of Mamba on unseen target domains.
> To address issue, the proposed method not only introduces both forward and backward directions to primarily preserve information about the most severe lesions, but also introduces a EM-based State Recalibration mechanism to compact the feature space so that the feature distribution is less sensitive to the domain shift.
>
> (2) From the experimental side, the proposed Samba shows a significant unseen domain performance improvement than the VMamba baseline on three different modalities, varying from fundus images, X-ray images, and pathological images, respectively.
> Moreover, as discussed in the paper, the recurrent encoding of image patches also contributes to cross-domain disease grading. In Tab.4, even the VMamba baseline outperforms CNN- and ViT-based methods.
>
> We will enrich these discussions accordingly.
>
> **Q2**: Ablation studies regarding the specific structure of Samba.
>
> **R:**
> We provide an ablation study on the specific structure of Samba.
> Specifically, two components, namely, Bi-directional State Space Modeling (BSSM) and EM-based State Recalibration (ESR),  are incorporated into the VMamba baseline.
> The experiments are conducted on the DG Fatigue Fracture Grading Benchmark, and the results are reported in Table R1 in the attached PDF.
> It is observed that BSSM contributes to an ACC, AUC and F1 improvement of 5.2\%, 1.7\% and 4.9\%, respectively.
> ESR contributes to an ACC, AUC and F1 improvement of 18.3\%, 9.4\% and 12.2\%, respectively.
>
> **Q3**: The open-source code is not explicitly provided.
>
> **R**: Thanks for your valuable suggestion. Owing to the company regulation, the source code will be made available after publication.
> We respectively ask for the reviewer's understanding on this restriction.
>
> **Q4**: Among the compared methods mentioned in Table 4, some ViT-based and ResNet-50-based methods are not specifically designed for domain adaptation tasks. Is this comparison fair?
>
> **R**: Thanks for your value comments, so that we could have a chance to clarify the compared methods.
> We first compare six methods for domain generalization tasks, namely, [36,50,51,53,55,56].
> Then, we compare two DR grading methods under the domain generalization setting, namely [4,8].
> All these eight methods are plenty for comparison under the context of domain generalization.
> The rest four CNN or ViT based methods [6,19,29,31] do not have domain generalization property, and are involved just for boarder comparison.
> Besides, these methods are implemented under the emperical risk minimization (ERM) setting, which is the same as the VMamba baseline.
> This demonstrates the effectiveness of Mamba structure in the cross-domain grading tasks.
>
> **Q5**: Can you explain how SSM selects severe lesion information?
>
> **R:**
> Thanks for your valuable suggestion.
>
> We model the relation between patch embedding from SSM and severity level by drawing inspiration from the class activation map (CAM) mechanism [a].
> Specifically, give an image $\boldsymbol{x}$, assume that $\boldsymbol{\tilde{f}}_k(i)$ denote the activation of unit $k$ from the last Samba block at the patch $i$.
> Then, for unit $k$, after implementing a global average pooling (GAP), the feature embedding $F^k$ is $\sum_i f_k(i)$.
>
> Thus, for a certain severity level $c$, the input to the softmax $S_c$ is $\sum_k w_k^c F_k$, where $w_k^c$ is the weight corresponding to severity level $c$ for unit $k$.
> Here $w_k^c$ indicates the importance of $F_k$ for severity level $c$.
> Finally, the output of the softmax for severity level $c$, $P_c$ is computed as
> $$
> P_c = \frac{{\rm exp} (S_c)}{\sum_c {\rm exp} (S_c)}.
> $$
> By plugging $F_k = P_i f_k(i)$ into the severity level score $S_c$, we obtain
> $$
> S_c = \sum_k w_k^c \sum_i f_k(i) = \sum_i \sum_k w_k^c f_k(i).
> $$
>
> Then, the severity activation map $M_c$ for severity level $c$ is defined as follows, where the activation of each patch $i$ is computed as
> $$
> M_c(i) = \sum_k w_k^c f_k(i).
> $$
>
> We take the patch embeddings from the last Samba block as input, so as to visualize the per-level severity activation patterns.
> Then, the activated severity patterns are displayed on the original images.
> We use FGADR as the unseen target domain.
>
> The results are shown in Fig.~R1 in the attached 1-pg PDF, where the activated patches are highlighted in blue boxes.
> From the first to the fifth row, the samples from level-1 to level-5 are provided accordingly.
> From the first to the fifth column, the patch activation map from level-1 to level-5 is generated by the aforementioned methods.
> Notice that, as level-1 refers to the normal scenario, each sample has activations on level-1, meaning some patches are normal.
>
> [a] Zhou, Bolei, et al. "Learning deep features for discriminative localization." Proceedings of the IEEE conference on computer vision and pattern recognition. 2016.
>
> **Q6**: In the comparative experiments with other SOTA methods, are all the source domains supplemented with the two additional large-scale datasets DDR [24] and EyePACS [13]?
>
> **R:** Yes, as all the performance of the state-of-the-art methods reported in [8] is supplemented with the two additional large-scale datasets DDR [24] and EyePACS [13], for fair evaluation, we also supplement both datasets when training.
> We will clarify it in the main text accordingly.
>
> Finally, should you have further suggestions and comments, we are glad to incorporate during the discussion stage.

---

> > ### Comment · Reviewer_b4sx · 2024-08-12
> >
> > Thanks to the authors for actively responding to the raised concerns and promising revisions in the updated version. However, considering the innovativeness of Samba (based on Mamba for various tasks), the score suitable for a Borderline Accept will not be altered.

---

> > > ### Author Response · Authors · 2024-08-12
> > > **Re: Official Comment by Reviewer b4sx**
> > >
> > > We are glad that you are satisfied with the revision and find the concerns resolved.
> > >
> > > We will improve our paper carefully per your valuable suggestions.

---

### Official Review · Reviewer_LrSF · 2024-07-09

**Soundness:** 2
**Presentation:** 3
**Contribution:** 2
**Rating:** 6
**Confidence:** 4

**Summary:**

The paper introduces a new method for disease grading for both within and cross-domain medical images. Three different imaging modalities have been used for experimentation including Retinal, X-ray, and H&E images. In terms of methodological novelty, the authors propose to encode image patches in a recurrent manner to capture informative lesions. Further, they use an EM-based recalibration method to reduce the cross-domain variance by compacting the feature space. Overall, their experiments shows that the method proposed is superior to the baselines.

**Strengths:**

The paper is well-written and has a strong theoretical base with detailed explanations. Besides the quantitative evaluations, the authors have conducted some qualitative experiments to show attention maps on retinal images. Also, the idea seems novel along with introducing new modules to tackle issues present in medical images datasets.

**Weaknesses:**

There are a few major issues with the experimental design:
1. The Cross-domain Breast Cancer Grading Benchmark is not a well-defined cross-domain problem in computational pathology. Firstly, this is not a widely used dataset in the field. Secondly, the images come from the same center yet with different scanning magnifications. It is not common to use images from two different magnifications as the "domains". The cross-domain problem in computational pathology is when the images are actually from two different centers and with two different staining protocols (images might be scanned in varying magnifications). An example of such a problem can be found here: https://camelyon16.grand-challenge.org/Data/

2. In Table 4, the authors have reported the benchmark from [8]. It has not been mentioned whether the proposed model was trained on the exact same seed, device, and cross-validation folds. If any of these are different, the comparison is not fair. Instead, I'd suggest the authors benchmark these with the same exact setting and report the results.

3. For highly imbalanced datasets, especially medical data, it is proper to report Balanced Accuracy to show the performance of the model on the rare classes. Both ACC and AUC fail to represent the rare classes. This is important for comparison as results in Table 1 show that F1 is significantly lower than ACC and AUC. Therefore, Balanced accuracy should be reported for all the benchmarks.

4. There are quite a few modules in different parts of the model that have not been studied in an ablation study. These modules are the bach norm, linear layer, etc. in the EM-based recalibration. Adding the ablation study for these modules can enhance the work :)

**Questions:**

I am curious if the authors can elaborate more on this conclusion made in page 15: "After processed by the Recurrent Patch Modeling module, more regions in the correlation The high-response patches have grade-related lesions and the information is transported in the recurrent process matrix have higher response."

**Limitations:**

It has been justified properly.

---

> ### Author Rebuttal · Authors · 2024-08-05
>
> **Q1**:  Is Cross-domain Breast Cancer Grading Benchmark proper? 1) not widely used. 2) It is not common to use images from two different magnifications as the "domains". 3) Evaluation on CAMELYON17, from different centers and different staining.
>
> **R**: Thanks for your valuable comment, so that we could have a chance to clarify some important aspects.
>
> 1) Following your suggestion, we take the five individual domains marked by the staining protocols in CAMELYON17 for further experiments.
> We use Domain-1 as the source domain, and reports its performance on the rest four unseen target domains, denoted from Domain-2 to Domain-5.
> The results are reported as follows.
> It is observed that, the proposed Samba still shows a significant performance improvement on all the four unseen target domains compared with the baseline VMamba-ERM.
> The results will be included when revision.
>
> Table R2: Generalization performance comparison between Samba and VMamba-ERM baseline.
> | Method | Domain-2 | Domain-3 | Domain-4 | Domain-5| avg.|
> |----------|----------|----------|----------|----------|----------|
> | VMamba-ERM  |  76.23  | 74.17 | 79.53 | 69.87 | 74.95 |
> | Samba |  **84.59**  | **82.48**  | **86.50** | **78.64** |  **83.05**  |
> ||||||
>
> 2) We also inspect in our dataset, if the samples vary in staining and if they have domain gap from a rigorous machine learning perspective.
> As shown in Fig.R2 a (attached in the 1-pg PDF), the samples from Domain-1 ($\times$20) and Domain-2 ($\times$40) are not only different in magnifications, but also varied in staining.
> Besides, we use t-SNE visualization to inspect the samples' feature distribution from Domain-1 and Domain-2, displayed in Fig. R2 b (attached in the 1-pg PDF), respectively.
> It can be observed that, samples from Domain-1 (marked in X) and Domain-2 (marked in O) have clear distance in the feature space. Besides, samples of the same severity level but from different domains are not clustered together. It indicates that the two domains in this dataset have the so-called domain gap and are suitable to benchmark the generalization capablity.
>
> 3) In real-world scenarios, some hospitals may use magnification factors that are not present in the training set. So testing the cross-magnification generalization ability by the Cross-domain Breast Cancer Grading Benchmarkalso still has practical significance.
>
> **Q2**: Are evaluation in Table 4 same and fair?
>
> **R**: We would like to clarify that, the proposed Samba is evaluated under all the default settings of [8] for fair evaluation.
> Specially, the batch size is 16, the training terminates after 100 epochs, the initial learning rate is $10^{-3}$ with a weight decay of $5\times10^{−4}$ and a momentum value of 0.9.
> We will explicitly mention these details.
>
> **Q3**: 1) Proper to report Balanced Accuracy, e.g., Table 1. 2) Balanced accuracy for all the benchmarks.
>
> **R**:
> 1) We adapt the balance accuracy metric (denoted as BACC), which is the mean of Sensitivity and Specificity, on DG Fatigue Fracture Grading benchmark (Table.R3) and DG Breast Cancer Grading Benchmark (Table.R4).
> The results are reported as follows.
> It can be seen that, the proposed Samba still outperforms the rest methods in term of the balance accuracy, indicating its effectiveness on rare classes.
>
> Table R3: Effectiveness of the proposed Samba on recurrent patch modeling under BACC metric. Domain-1 and Domain-2 in the Fatigue Fracture Grading
> Benchmark are used as the source and unseen target domain.
> | Method | ACC | AUC | F1 | BACC | |
> |----------|----------|----------|----------|----------|----------|
> | LSTM | 39.8 | 50.2 |18.6 | 25.6 |
> | UR-LSTM | 43.3 | 61.8 | 20.9 | 25.2 |
> | UR-GRU | 45.7 | 65.1 | 22.4 | 27.1 |
> | ViT  | 50.0 | 69.3 | 26.5  | 30.9 |
> | VMamba-ERM  | 52.7 | 70.4 | 28.7 | 34.0  |
> | Samba  |  **76.2** | **81.5** | **45.8** | **52.2**  |
> ||||||
>
> Table R4: Effectiveness of the proposed Samba than baseline under BACC metric. Experiments on DG Breast Cancer Grading Benchmark.
> | Method | Backbone | ACC | BACC | ||
> |----------|----------|----------|----------|----------|----------|
> | ERM | VMamba-T | 40.4 | 18.7 |  |
> | Samba | VMamba-T | **54.8** | **24.5** |  |
> | ERM | VMamba-S | 50.1 | 20.6 |  |
> | Samba  | VMamba-S | **56.1** | **27.9**  |  |
> | ERM  | VMamba-B | 54.9 | 25.6 |  |
> | Samba |  VMamba-B | **60.5** | **29.8** |  |
> ||||||
>
> 2) We would like to raise the reviewer's attention that, despite that the severity level is highly imbalanced in grading problems especially in DR grading, ACC, F1 and AUC metrics are both acknowledged as the commonly-used evaluation metrics [4,6,8,19,29,36,50,51,53,54,55]. Therefore, we directly adapt these metrics and existing evaluation protocols for fair evaluation in Table 4.
> Besides, balanced accuracy probes a model's performance very similar as F1-score does, while we already use F1-score.
>
> **Q4**: Ablation on mdules.
>
> **R**: We provide an ablation study on each module of the proposed Samba, which is also mentioned by Reviewer\#Wyiu and b4sx.
> Specifically, on top of the VMamba baseline, two components, namely, Bi-directional State Space Modeling (BSSM) and EM-based State Recalibration (ESR), are added.
> The experiments are conducted on the DG Fatigue Fracture Grading Benchmark. The results are reported in Table~R1 in the attached 1-pg PDF file.
> It is observed that BSSM contributes to an ACC, AUC and F1 improvement of 5.2\%, 1.7\% and 4.9\%, respectively.
> ESR contributes to an ACC, AUC and F1 improvement of 18.3\%, 9.4\% and 12.2\%, respectively.
> Meanwhile, we respectively ask for the reviewer's understanding that our work focuses on innovatively introducing BSSM and ESR on top of the vanilla elements (e.g., layer, norm) in VMamba without modify. The ablation of vanilla elements may beyond the scope of this work.
>
> **Q5**: Elaborate more on text in p15.
>
> **R**: Please refer to the general response.
>
> Should you have further suggestions, we are glad to address during the discussion stage.

---

> > ### Comment · Reviewer_LrSF · 2024-08-09
> >
> > I would like to thank the authors for their significant effort and for providing more details in the rebuttal phase.
> >
> > However, I believe the Breast Cancer Grading Benchmark is not a representative choice for claiming that the proposed approach is a good solution for a computational pathology task. Multiple well-studied and standard datasets in the field, such as TCGA sub-datasets, could have been chosen. These datasets can be used for grading (to match the rest of the paper) or subtyping tasks based on the proposed research. Camelyon17 is also a viable choice if the authors decide to work on subtyping.
> >
> > Another important factor is that several ablation studies have been conducted on the Breast Cancer Grading Benchmark, which is still questionable in terms of reliability and generalizability of results. This dataset should ideally be replaced with another dataset, as mentioned above.
> >
> > I also appreciate the authors' effort in providing details in the PDF file. It is important to note that there are stain variations within the same center samples due to the staining procedure, which accounts for in-domain data variance. Additionally, differences in magnification are not technically a domain-shift problem; rather, they are known as a cross-scale problem [1][2], which is not aligned with the paper's topic and the rest of the experiments. Thus, to fairly test the hypothesis in this field, the study should cite and compare proper literature on a standard dataset.
> >
> > I have also considered the paper that released the Breast Cancer Grading Benchmark [3] dataset. However, within that paper, the authors do not consider their dataset as a cross-domain dataset and instead build a cross-scale model.
> >
> > Based on the above rationale, I am not convinced to change my rating.
> >
> > [1] Sikaroudi, M., Ghojogh, B., Karray, F., Crowley, M., and Tizhoosh, H.R., 2021, April. Magnification generalization for histopathology image embedding. In 2021 IEEE 18th International Symposium on Biomedical Imaging (ISBI) (pp. 1864-1868). IEEE.
> >
> > [2] Chhipa, P.C., Upadhyay, R., Pihlgren, G.G., Saini, R., Uchida, S., and Liwicki, M., 2023. Magnification prior: a self-supervised method for learning representations on breast cancer histopathological images. In Proceedings of the IEEE/CVF Winter Conference on Applications of Computer Vision (pp. 2717-2727).
> >
> > [3] Yan R, Ren F, Li J, Rao X, Lv Z, Zheng C, Zhang F. Nuclei-Guided Network for Breast Cancer Grading in HE-Stained Pathological Images. Sensors. 2022; 22(11):4061.

---

> ### Author Response · Authors · 2024-08-11
> **Reply to Q1: Effectiveness on a common computational pathology dataset.**
>
> **Q1**: Effectiveness on a common computational pathology dataset.
>
> **R**: We respect and value the reviewer’s perspective from computational pathology.
> We adopted CAMELYON17 to conduct the two experiments in this paper where Cross-domain Breast Cancer Grading Benchmark has been used for validation.
> Same as the earlier rebuttal, five individual domains marked by the staining protocols and sites in CAMELYON17 are used for the cross-domain experiments. We use Domain-1 as the source domain, and reports its performance on the rest four unseen target domains, i.e., from Domain-2 to Domain-5.
>
> The first experiment (Table 2 in the main text) is the impact of the number of components K in GMM, where ACC, AUC and F1 are used as evaluation metric. The performance on CAMELYON17 is attached as follows.
>
> Table 2: Impact of the number of components K in GMM on unseen target domain performance. Experiments are conducted on the CAMELYON17. Domain-1 is used as source
> domain. Metrics presented in percentage (%).
>
> | | | Domain-2 | | | Domain-3 | | | Domain-4 | | | Domain-5 | |
> |----------|----------|----------|----------|----------|----------|----------|----------|----------|----------|----------|----------|----------|
> | K value | ACC | AUC | F1 | ACC | AUC | F1 | ACC | AUC | F1 | ACC | AUC | F1 |
> | 16 | 81.80 | 92.09 | 79.96 | 79.45 | 88.96 | 76.60 | 83.92 | 95.38 | 82.10 | 75.84 | 81.99 | 74.16 |
> | 32 | 82.68 | 93.81 | 80.95 | 80.39 | 91.28 | 77.97 | 84.38 | 96.20 | 82.96 | 76.53 | 82.46 | 74.92 |
> | 48 | 84.06 | 94.25 | 81.83 | 81.84 | 92.60 | 78.49 | 85.87 | 97.16 | 83.87 | 77.69 | 83.85 | 75.04 |
> | 64 | **84.59** | **95.67** | **83.10** | **82.48** | **93.32** | **79.85** | **86.50** | **97.82** | **85.13** | **78.64** | **84.70** | **75.32** |
> | 96 | 84.27 | 95.48 | 82.53 | 82.06 | 92.90 | 79.44 | 86.16 | 97.45 | 84.80 | 78.15 | 84.28 | 74.86 |
> | 128 | 83.65 | 94.70 | 81.97 | 81.50 | 92.41 | 78.36 | 85.47 | 96.90 | 84.62 | 77.38 | 83.66 | 74.14 |
> ||||||||||||||
>
> Default K is set to 64, and we further test the performance when K is set to 16, 32, 48 and 96, respectively.
> The results are reported in Table 2. When K is set to 64, the proposed Samba achieves the best grading performance.
> This observation is consistent to the performance on Cross-domain Breast Cancer Grading Benchmark, where a number of 64 Gaussians achieves the optimal performance.
>
> The second experiment (Table 3 in the main text) is to analyze the computational cost and performance trade between the VMamba-ERM and the proposed Samba, where only accuracy is used as the evaluation metric. The performance on CAMELYON17 is attached as follows.
>
> Table 3: Computational cost comparison between VMamba-ERM and the proposed Samba. Experiments are conducted CAMELYON17. Domain-1 is used as source domain. Metrics in percentage (%).
>
> |Method | Backbone | GFLOPS | Para. | Domain-2 | Domain-3 | Domain-4 | Domain-5 | avg. |
> |----------|----------|----------|----------|----------|----------|----------|----------|----------|
> | EMR | VMamba-T | 3.7 | 32.7 | 70.08 | 67.29 | 72.96 | 63.16 | 68.37 |
> | Samba | VMamba-T | 5.5 | 32.7 | **78.74** | **76.15** | **80.06** | **71.05** | **76.50** |
> | EMR | VMamba-S | 7.9 | 63.4 | 72.86 | 69.50 | 75.08 | 65.72 | 70.79 |
> | Samba | VMamba-S | 11.3 | 63.4 | **81.01** | **78.96** | **82.75** | **73.88** | **79.15** |
> | EMR | VMamba-B | 14.0 | 112.4 | 76.23 | 74.17 | 79.53 | 69.87 | 74.95 |
> | Samba | VMamba-B | 19.6 | 112.4 | **84.59** | **82.48** | **86.50** | **78.64** | **83.05** |
> ||||||||||
>
> The same trend is observed on this dataset, where using Samba on each type of the VMamba backbone shows a clear performance improvement on unseen domains.
>
> We sincerely wait for the reviewer’s feedback about the above experiments, which function the same as the Cross-domain Breast Cancer Grading Benchmark in the submission.
> If they could meet the reviewer’s standard on computational pathology, we are happy and open to incorporate it into our work, subtyping the computational pathology part.

---

> ### Author Response · Authors · 2024-08-11
> **Reply to Q2: Cross-scale or Cross-domain?**
>
> **Q2**: Cross-scale problem or Cross-domain problem?
>
> **R**:
> First of all, we would like to thank the reviewer for the valuable feedback on the *cross-scale* perspective.
>
> However, we humbly suggest the reviewer that there might be some misunderstanding on *cross-domain*.
>
> From a machine learning perspective, where the venue of NeurIPS focuses more on, as long as the source and unseen target domains are *not independent and identically distributed* / *not the same* [a,b,c], the domain gap exists and the domain generalization techniques can be applied.
> The change of scale, caused by the magnification, also poses distribution shift between the source and target domain, can also be treated as a type of domain generalization, according to this machine learning definition.
>
> [a] Zhou, Kaiyang, et al. "Domain generalization: A survey." IEEE Transactions on Pattern Analysis and Machine Intelligence 45.4 (2022): 4396-4415.
>
> [b] Zhou, Kaiyang, et al. "Domain Generalization with MixStyle." International Conference on Learning Representations. 2021.
>
> [c] Wang, Jindong, et al. "Generalizing to unseen domains: A survey on domain generalization." IEEE transactions on knowledge and data engineering 35.8 (2022): 8052-8072.
>
> Therefore, we humbly suggest that conceptually from a machine learning perspective, the Cross-domain Breast Cancer Grading Benchmark is applicable to validate the generalization capacity of a model.
> Besides, as we have already provided in the 1-pg PDF file, the distribution shift exists between the Doman-1 and Domain-2 of this dataset.
> This clearly-existed domain gap, along with the definition of domain generalization, means the Cross-domain Breast Cancer Grading Benchmark is rational to benchmark the domain shift of grading problems.
> In real-world scenarios, some hospitals may use magnification factors that are not presented in the training set. Hence, testing the cross-magnification generalization ability by the Cross-domain Breast Cancer Grading Benchmark still has the practical significance.
>
> **Summary**:
> As we mentioned at the beginning, we respect and value the reviewer’s perspective from computational pathology.
> We are glad and open if the reviewer feel it would be better to treat the results on cross-domain breast cancer grading are a discussion on the scale generalization or replace them by the results on CAMELYON17.
>
> We hope we can reach a consensus, and look forward to your feedback and suggestions.

---

> > ### Comment · Reviewer_LrSF · 2024-08-12
> >
> > I appreciate the detailed response and the effort of the authors to provide further insights.
> > I understand that DG on cross-scale data is a valid problem from an ML point of view. Yet, my main concern is rooted in the choice of the dataset, where among many well-studied datasets present, the BCGR dataset had been chosen. Still, I am not convinced it is a good representative unless an extensive benchmark with different methods is provided.
> >
> > However, by including C17, and providing the ablation studies and the benchmarking results on that, my initial concern was addressed. With that, I strongly suggest the author use the C17 as the main pathology representative dataset and use the BCGR results as a secondary dataset. Also, a clarification section needs to be added to the work to explain that BCGR is a cross-scale generalization and is not multi-domain as it is known in pathology. This will ensure the reliability of the result for a pathology-related audience. Given the new set of results, I would raise my rating to a weak accept :)

---

> ### Author Response · Authors · 2024-08-12
> **Re: Official Comment by Reviewer LrSF**
>
> We are glad that your concerns have been properly addressed.
>
> We will significantly polish and improve our work per you suggestions, particularly on prioritizing C17 over BCGR for a professional standard of the pathology perspective.
>
> Finally, thanks again for your time and effort in helping us improve our work.

---

### Official Review · Reviewer_Wyiu · 2024-07-12

**Soundness:** 2
**Presentation:** 2
**Contribution:** 3
**Rating:** 5
**Confidence:** 4

**Summary:**

The authors propose a model which they call Samba, for Severity-aware recurrent modelling, which is a method designed for cross-domain medical image grading. They introduce several challenges in medical grading, namely the difficulty models encounter in generalising to unseen domains, as well as the existence of ambiguity in lesion severity grading. The model is comprised of two main blocks: recurrent bidirectional Vision Mamba layers and a Expectation-Maximisation State Recalibration (EMSB) module which consists of learnable tokens which capture lesion representations, and which are then used as bases to map to more compact feature embeddings. These are then used to initialise an Expectation-Maximisation (EM) algorithm and the lesion feature distribution is estimated using Gaussian Mixture Models (GMMs) for each image. The Mamba layers treat the image patches as sequential data, with the rational being the relevant information will be propagated through the hidden states. The EM module models the feature distribution of lesions using the GMMs to try and eliminate domain shift. Samba is applied to three benchmarks, where images are separated between a source domain and a target domain: Diabetic Retinopathy fundus images (DR), Fatigue Fracture X-rays and Breast Cancer histopathology images. The model is trained on the source domain and applied to the target domains. Ablation is carried out, comparing Samba to different baseline and SOTA models, as well as looking at iteration number and severity base update methods. The results obtained show Samba generally outperforms existing methods to some degree. The authors also provide a theoretical analysis on the generalisation risk bound in the Appendix.

**Strengths:**

The paper gives a good introduction to the challenges inherent in grading disease severity in the medical imaging domain. It tackles an important problem, as we know current algorithms often fail at generalising across domain. Integrating Vision Mamba layers with EM-based recalibration of image features is a nice contribution designed to tackle this challenge, which is motivated by logical reasoning about the properties of medical images. The method is comprehensively evaluates on three different benchmark datasets, which are specifically designed to test domain generalisation of algorithms. Furthermore, the results suggests that Samba is able to generalise to unseen domains better than other established methods. There is also ablation provided on several aspect of the model (number of components K, iteration T and update on $\mu$), as well as t-SNE plots showing how the model embeddings cluster better by severity grading and not data provenance, as well as attention maps of DR images. Overall, the paper combines ideas from current SOTA in deep learning (mamba), statistical modelling (GMM-EM) and properties of medical images.

**Weaknesses:**

In my opinion the paper's main weaknesses are the following:

1 - Unclear explanation of the EM-based State Recalibration module

I think the introduction to the EM-based State Recalibration module in this paper is too brief, perhaps assuming reader familiarity, and does not provide intuition as to why using GMMs + EM makes sense in this context or is more suitable to this task than other potential approaches. The implementation details of how the the GMM \& EM is integrated into the overall Samba architecture are not clearly or fully presented. The initialisation strategy for the EM is unclear to me: the paper mentions using learnable severity bases to initialise EM, but doesn't explain how these bases are learned or why this initialisation is most beneficial. It also doesn't discuss the convergence criteria used for the EM and although it shows ablation into how many iterations are best, this comes without proper introduction in the background or methods sections. The same goes for the number of Gaussian basis used: you show ablation on this, but don't introduce this in background or methods sections. Finally, although some equations are presented outlining the E- and M-steps, variables are not always defined and the choice of kernel function seems entirely ad-hoc and unjustified. In particular

You define your GMM as:

$$p(f_n) = \sum_{k=1}^K z_{nk} \mathcal{N}(f_n|\mu_k, \Sigma_k),$$

but $z_{nk}$ typically represents responsibilities which are estimated in the E-step, not given as part of the model definition, where $\pi_k$ would represent the mixing coefficients of the GMM. Then you derive the E-step as:

$$z_{nk} = \frac{\mathcal{K}(f_n, \mu_k)}{\sum_{i=1}^K \mathcal{K}(f_n, \mu_i)}$$

whereas a standard formulation would be

$$z_{nk} = \frac{\pi_k \mathcal{N}(f_n|\mu_k, \Sigma_k)}{\sum_{i=1}^K \pi_i \mathcal{N}(f_n|\mu_i, \Sigma_i)}.$$

However, you don't justify the use of the arbitrary kernel function $\mathcal{K}$ (exponential inner dot $\mathrm{exp}(f^T\mu)$ over standard Gaussian probabilities. Could you explain this in more detail? Then you don't explicitly define what $Z^t$ represents, nor what the relationship is with $z_nk$. Likewise with $F$, you do not explicitly define what it represents, but I will assume you mean the feature matrix of the input image. How does $F$ relate to equation (5)? How does this derive into equation (6)? The recalibration step $\tilde F$ is not well explained in terms of matrix operations or dimensions. This lack of clarity makes it difficult to interpret the method in terms of a well-established optimisation algorithm, which is an important issue as explaining the transformations applied to the feature representations is crucial to understanding Samba. Overall, while the general idea of using GMM-EM for feature recalibration is interesting, the mathematical formulation presented in the paper has inconsistencies and deviations from standard GMM-EM that I don't think are well-justified and which I would like to see more fully explained and motivated.

2 - Unclear structure of results and lack of clear baseline comparisons

While the paper compares three datasets to various baseline and baseline methods, each dataset is treated separately, so it's not always clear the results shown extend to the other datasets, if the comparisons are fair or how the baselines were implemented. I feel like the main results are the ones presented for the Diabetic Retinopathy dataset, yet these are presented last. I think the results section would read better and make more sense if these were presented first. However, one of my main issues here is that you only mention your main baseline comparison method on p.7, but you compare against in Figure 3 without even mentioning it. This should be properly introduced and you should explain why VMamba under empirical risk minimisation (VMamba-ERM) is a good baseline model against which to compare. You also need to explain how Samba differs from VMamba-ERM and what advantages these differences bring to the task that the other doesn't provide. Finally, you don't provide ablation on Samba itself, showing the individual contribution of the Mamba layers vs EMBS layers. With regards to the theoretical analysis on the generalisation risk bound, it feels disconnected to the rest of the paper and not particularly relevant to the stated aims. Is there some way to tie it more or explain why its important? Finally, I think an Appendix section on how the datasets were preprocessed and showing the hyperparameters employed for all the other models presented in the results sections would enrich this paper. There are other things which I find unclear, which I have put below in Questions.

**Questions:**

Line 44 - I don't fully understand this sentence. Maybe you could rephrase?

Line 54 - at the distal of what?

Line 123 - what's a stem unit?

Figure 2 - I think you need to describe the Samba blocks in more detail - here it looks like the bi-directional SSM and EM recalibration occurs in the downsampling block. Is this a mistake? What'a VSS block?
What's the difference between the training and inference arrows in the figure?

Figure 3 - Iteration Number T - What models are you comparing yours against? What is Vmamba-ERM and what is SOSS?? Why is Vmamba showing a constant trend across iteration number T? What are the different columns?

Line 235 - why does the number of iterations play an important role in the EM algorithm? You should introduce this beforehand and the ablation study should answer this question.

Line 237 - do you mean Figure 3 instead of Figure 6 here? - Figure 6 shows the effect of regularisation techniques and is situated in the appendix.

Line 244 - why are these these methods appropriate to test different optimisation techniques here? Please explain this.

Line 252 - these are percentage points, not percentage of improvement.

Table 1- Why is Table 1 showing three F1 scores highlighted in brighter green? and two blocks of yellow and light green? Is this a mistake? Also what do these colours represent?

Table 3 - OK, why do you think Samba is doing better there?

Line 283 - percent points, not percents...

Table 4 - why do you think Samba and Vmamba-ERM are doing so well on FGADR compared to other methods?

A.3 - the text needs a rewrite to make sense I think. Please check this.

Figure 8/9 - Do you have any ground truths available to compare the attention maps shown in Figure 8 and 9?

**Limitations:**

The authors point out some limitations of their work with regards to class imbalance. I have pointed out other perceived limitations in the Weakness section above.

---

> ### Author Rebuttal · Authors · 2024-08-05
>
> **Q1**: Unclear explanation. **R**:
>
> 1) After proceeded by BSSM, each patch embedding $\boldsymbol{f}_n$ is projected by a linear layer to get the state embedding $\boldsymbol{x}_n$. The severity base $\boldsymbol{\mu}_n$ is modeled as a mixture of Gaussian from $\boldsymbol{x}_n$.
> The E-M algorithm iteratively approximates $\boldsymbol{\mu}_n$ to $\boldsymbol{x}_n$.
> After convergence, the embedding is fed into the rest module.
> 2) As introduced in L194-196, moving average is adapted to update the $\boldsymbol{\mu}^{0}$, which is used as the initial Gaussian parameter in each EM process.
> 3) The E-M empirically implements 3 iterations.
> 4) We will define the number of Gaussian kernels $K$ in L171.
> 5) We will mention that the mixing coefficients of GMM are left out for simplicity and easy computation and the exponential inner dot kernel is used.
> 6) In the $t$-th iteration, $\mathbf{Z}^t$: the responsibilities of all the patch embeddings from a sample, where $\mathbf{Z}^t=\{z_{nk}^t\}$.
> 7) Typo. L185, remove $\mathbf{F}$.
> 8) In Eq.6, $\mathbf{\mu}$: the Gaussian basis of all the patch embeddings from a sample, where $\boldsymbol{\mu}=\{\mu_{k}\}$. $\mathbf{\widetilde{F}}$: the entire image feature from all the patch embeddings.
>
> **Q2**: Structure.  **R**:
>
> 1) To clarify, as this paper handles cross-domain medical grading, each dataset we use has more than one domains.
> The evaluation of domain generalization is evaluated inside each dataset, not across different datasets.
> Tab.4: pls refer to general response.
>
> 2) We will put Table 4 and the subsection first.
>
> 3) We will metion before Fig.3. Especially,
> The comparison is made between the proposed Samba and the vanilla VMamba as baseline.
> The VMamba baseline is trained under the empirical risk minimization (ERM), which we denote as Vmamba-ERM.
> Notice that, training under ERM is a common baseline for domain generalization.
> On top of Vmaba-ERM, the advancement of Samba is two modules EMSR and BSSM.
>
> 4) Per-component ablation: in general response.
>
> 5) We will put the theoretical analysis to the supplementary, making room for details of the method and ablation study.
>
> 6) Details of data pre-processing and hyper-parameter settings on each dataset will be in Appendix.
>
> **Q3**: rephase L44. **R**: *Domain generalized disease grading aims to learn a model that can be well generalized to unseen target domains, when the model is only trained on the source domain data.
> Practically, the feature distribution between the source and unseen target domains usually varies.*
>
> **Q4**: L54.
> **R**: *at the distal of the blood vessels.*
>
> **Q5**: Stem unit?
> **R**: A stem unit partitions the input image into patches.
>
> **Q6**: Typo in Fig.2. **R**:
>
> 1) Both modules should occur in the Samba block. Will correct.
>
> 2) Typo. 'VSS block'->'Samba block'.
>
> 3) Training and inference input images from different domains, marked by red and green arrows, respectively. We will distinguish the source and target domains (located at the upper-left of Fig.2) using different colors.
>
> **Q7**: Fig.3. compare what? What's Vmamba-ERM\&SOSS? Why Vmamba is constant across T? Different columns? **R**:
>
> 1) The comparison is made between the proposed Samba and the vanilla VMamba as baseline, which is trained under the empirical risk minimization (ERM) and denoted as Vmamba-ERM.
>
> 2) Typo. 'SOSS' -> 'Samba'.
>
> 3) VMamba-ERM does not have EM-based State Recalibration. EMSR is parameterized by T.
> Therefore, the performance of Vmamba-ERM is consistent to T.
>
> 4) From column 1 to 3, it reports the AUC, ACC and F1 metric.
>
> **Q8**: Why iteration number?
> **R**: E-M algorithm implements the approximation by iteratively conduct the E and M step [12].
> A small iteration number does not reach the convergence criterion, which results in an poor approximation.
> After a too-large iteration number, the approximation may already reach the optimal, adding unnecessary computational cost and training time. We will introduce \& clarify.
>
> **Q9**: Line 237.
> **R**: Typo. Fig.6->Fig.3.
>
> **Q10**: L244-explain why proper?
> **R**: The process of the proposed state recalibration is differentiable, thereby enabling the application of back-propagation to update $\boldsymbol{\mu}^0$. However, the stability of the update cannot be guaranteed due to the EM iterations. Therefore, we adopt moving average to update $\boldsymbol{\mu}^0$ to avoid collapse.
>
> **Q11\&14**: L252&283 - typos on percentage points.
> **R**: will correct.
>
> **Q12**: Tab.1
> **R**: Brighter green, yellow and light green intend to highlight the best, second and third top performance. We will correct.
>
> **Q13**: Tab.3, why Samba better?
> **R**: Compared to VMamba-ERM baseline, the EM-based State Recalibration in Samba models the feature distribution of lesions via Gaussian Mixture Model with learnable severity bases, and re-estimates by E-M algorithm. The grading-related features are mapped to a more compact space. It can be more stable in unseen target domains.
>
> **Q15**: Why much better on FGADR?
> **R**: FGDAR has a different severity-level sample distribution than other datasets.
> The samples without DR (level-1) only occupy only 5.5\% among all the training samples, which is far less than others (e.g., level-1 samples occupy 49.3\% in APTOS).
> Therefore, other methods may over-fit other severity levels and under-fit level-1.
> In contrast, the selective scan mechanism of the Vmamba-ERM and Samba allows the this severity distribution shift.
> The EM state re-calibration in Samba makes the feature space more compact, and improves the generalization.
>
> **Q17**: Fig.8/9. ground truth?
> **R:** As the datasets are for grading, we respectfully ask for the reviewer's understanding, there are no available fine-grained ground truth (e.g., pixel, boxes).
>
> Finally, should you have further questions or suggestions, we are happy to address during the discussion stage.

---

> > ### Comment · Reviewer_Wyiu · 2024-08-12
> > **Response to Author's Rebuttal**
> >
> > Thanks to the authors for the time and effort put into the rebuttal. I have read their response, as well as the other reviews and their associated comments.
> >
> > I agree with the authors that testing across multiple magnification (i.e. scales), although not the standard approach, can constitute cross-domain adaptation if it can be shown there exists a clear difference in the data distribution across domain. They have shown this to be the case with their UMAP embeddings and changes in magnification and resolution across scanners resulting in models underperforming is a know phenomenon in computational pathology. Additionally, they also test on X-ray fatigue fractures and Diabetic Retinopathy datasets.
> >
> > I appreciate the authors promise to update the structure of the paper: modifying/correcting pipeline figure, expanding on background and motivation, clarifying model explanation and mathematical formulation in 3.3, moving 3.4 to Appendix, introducing baseline models clearly, introducing first results with Figure 4 (DR dataset), correcting Table 1 and expanding on model ablation as shown in Table R1. Given the above reasons and the substantial work required to put these into effect, I am maintaining my original rating.

---

> > > ### Author Response · Authors · 2024-08-12
> > > **Re: Response to Author's Rebuttal**
> > >
> > > We are glad to see your concerns have been addressed.
> > >
> > > We will improve our paper carefully according to your valuable suggestions.

---

### Author Rebuttal · Authors · 2024-08-05

General Response:

We thank the reviewers for their time and constructive suggestions, and are glad that the reviewers unanimously give appreciation in a few points:

- Technique contribution (**Wyiu**: integrating Vision Mamba layers with EM-based recalibration of image features is a nice contribution; **LrSF**: the idea seems novel; **b4sx**: use state space model to store and transmit severity information from local to global; **hFq3**: significant problem.)

- Extensive evaluation \& visualization (**Wyiu**: comprehensively evaluation; **LrSF**: quantitative evaluations and some qualitative experiments; **b4sx**: The interpretability visualization of the experimental results is excellent; **hFq3**: different diseases and imaging modalities.)

- Motivation/Task significance (**Wyiu**: generalising across domain; **LrSF**: has a strong theoretical base with detailed explanation;  **hFq3**: significant problem.)

However, there are also some major concerns as follows, in which we clarify as follows.

- Lack of per-component ablation study (**Wyiu** comment\#2; **LrSF** comment\#4; **b4sx** comment\#2).

**R**: We further provide an ablation study on each module of the proposed Samba.
Specifically, on top of the VMamba baseline, two key components, namely, Bi-directional State Space Modeling (BSSM) and EM-based State Recalibration (ESR), are evaluated.
The experiments are conducted on the DG Fatigue Fracture Grading Benchmark, and the results are reported in Table~R1 (attached in the 1-pg PDF file).
It is observed that BSSM contributes to an ACC, AUC and F1 improvement of 5.2\%, 1.7\% and 4.9\%, respectively.
ESR contributes to an ACC, AUC and F1 improvement of 18.3\%, 9.4\% and 12.2\%, respectively.

- Path to visualize and understand the severity level (**b4sx** Comment\#5; **hFq3**: comment\#3).

**R**: We model the relation between patch embedding from SSM and severity level by drawing inspiration from the class activation map (CAM) mechanism [a].
Specifically, we take the patch embeddings from the last Samba block as input to generate the per-level severity activation patterns.
Then, the activated severity patterns are displayed on the original images.
We use FGADR as the unseen target domain.
The results are shown in Fig.~R1 (attached in the 1-pg PDF file), where the activated patches are highlighted in blue boxes.
From the first to the fifth row, the samples from level-1 to level-5 are provided accordingly.
From the first to the fifth column, the patch activation map from level-1 to level-5 is generated by the aforementioned methods.
Notice that, as level-1 refers to the normal scenario, each sample has activations on level-1, meaning some patches are normal.

[a] Zhou, Bolei, et al. "Learning deep features for discriminative localization." Proceedings of the IEEE conference on computer vision and pattern recognition. 2016.

- Clarify if the state-of-the-art comparison (Table 4) is fair (**Wyiu** comment\#2; **LrSF** comment\#2; **b4sx** Comment\#4).

**Wyiu**, **LrSF**, **b4sx**: For Table 4, the results of the state-of-the-art methods are directly cited from [8], and the evaluation protocols of the proposed method follow all the default settings as [8] does.

**LrSF**: Fair mplementation details. Specially, the batch size is 16, the training terminates after 100 epochs, the initial learning rate is $10^{-3}$ with a weight decay of $5\times10^{−4}$ and a momentum value of 0.9.

- A.3. The text needs a rewrite (**Wyiu** comment\#16; **LrSF** comment\#5).

**R:** The sentences have been re-written as:
*After processed by the Recurrent Patch Modeling module, more cells in the correlation matrix have higher response.
Usually, a handful of the patches inside the image have grade-related lesions.
After the processing of our module, the information of these grade-related lesions is transported to other patches. It allows the model to perceive a more global-wise representation.
Consequently, more patches that contain the grade-related lesion information are activated, and more cells are highly responded in the correlation matrix.*

We will explicitly mention these details when revision.

We hope our clarification could help to make a more informed evaluation to our work.
In the following individual response, we provide answers to each raised weakness/question.

Best regards,

Authors

---

### Decision · Program_Chairs · 2024-09-25

**Decision:**

Accept (poster)

**Comment:**

The paper was reviewed by four experts. Although there was some disagreement about the novelties and clarity of the paper before the rebuttal and discussion stages, the authors were able to address most of the concerns. All reviewers recommend acceptance of the paper.  The authors are advised to address the requested clarifications in the final version of the paper.